# FACTOR GRAPH OPTIMIZATION FOR BELIEF PROPAGATION DECODING

## ABSTRACT

Belief Propagation (BP) is a highly efficient message-passing algorithm for inference on graphical models, famously applied to the decoding of sparse codes. The performance of BP, however, is critically dependent on the structure of the underlying factor graph. Designing a graph structure that is optimal for BP decoding remains a significant challenge, especially when constrained by short block lengths or novel channel models. In this work, we introduce, for the first time, a gradient-based and data-driven framework to directly optimize the factor graph for the Belief Propagation algorithm. We learn locally optimal graph structures by running simulations under channel noise. This is enabled by a novel, complete graph tensor representation of the Belief Propagation algorithm, which makes the decoding process end-to-end differentiable. This representation allows us to optimize the graph structure over finite fields via backpropagation, coupled with an efficient line-search method. When applied to the design of sparse codes, the resulting BP-optimized factor graphs demonstrate decoding performance that outperforms existing popular codes and show the power of data-driven approaches for code design.

## 1 INTRODUCTION

Reliable digital communication is of major importance in the modern information age and involves the design of codes that can be robustly and efficiently decoded despite noisy transmission channels. Over the last half-century, significant research has been dedicated to the study of capacity-approaching Error Correcting Codes (ECC) (Shannon, 1948). Despite the initial focus on short and medium-length linear block codes (Berlekamp, 1974), the development of long channel codes (Forney, 1966; Costello & Forney, 2007) has emerged as a viable approach to approaching channel capacity (Berrou et al., 1993; MacKay, 1999; Richardson et al., 2001; Richardson & Urbanke, 2001; Arikan, 2008; Luby et al., 2001; Kudekar et al., 2011).

While the NP-hard maximum likelihood rule defines the target decoding of a given code, developing more practical solutions generally relies on theories grounded upon asymptotic analysis over conventional communication channels. However, modern communication systems rely on the design of short and medium-block-length codes (Liva et al., 2016) and the latest communication settings provide new types of channels. This is mainly due to emergent applications in the modern wireless realm requiring the transmission of short data units, such as remote command links, Internet of Things, and messaging services (De Cola et al., 2011; Boccardi et al., 2014; Paolini et al., 2015; Durisi et al., 2016; ESTI, 2021). These challenges call for the formulation of data-driven solutions, capable of adapting to various settings of interest and constraints, generally uncharted by existing theories.

The vast majority of existing machine-learning solutions to the ECC problem concentrate on the design of *neural decoders*. The first neural models focused on the implementation of parameterized versions of the legacy Belief Propagation (BP) decoder (Nachmani et al., 2016; 2018; Lugosch & Gross, 2017; Nachmani & Wolf, 2019; Buchberger et al., 2020). Recently, state-of-the-art learning-based de novo decoders have been introduced, borrowing from well-proven architectures from other domains. A Transformer-based decoder that incorporates the code into the architecture has been recently proposed by (Choukroun & Wolf, 2022a), outperforming existing methods by sizable margins and at a fraction of their time complexity. This architecture has been subsequently integrated

into a denoising diffusion models paradigm, further improving results (Choukroun & Wolf, 2022b). Subsequently, a universal neural decoder has been proposed in (Choukroun & Wolf, 2024b), capable of unified decoding of codes from different families, lengths, and rates. Most recently and related to our work, (Choukroun & Wolf, 2024a) developed an end-to-end learning framework capable of co-learning binary linear block codes along with the neural decoder.

However, neural decoding methods require increased computational and memory complexity compared to their well-established classical counterparts. Due to these challenges, and the non-trivial acceleration and implementation required, neural decoders were never deployed in real-world systems, as far as we know.

In this work, given the ubiquity and advantages of the Belief Propagation (BP) algorithm (Pearl, 1988; Richardson et al., 2001) for sparse codes, we consider the optimization (with respect to the channel capacity) of codes with respect to BP via the learning of the underlying factor/Tanner graph. From a graphical probabilistic model perspective (Koller & Friedman, 2009), BP being a marginalization algorithm, a gradient-based of a score metric method is given for the *structure learning* of BP's underlying Bayesian network in an end-to-end fashion. As far as we can ascertain, this is the first time a gradient-based data-driven solution is given for the design of the codes themselves based on a classical decoder. Such a solution induces a very low overhead (if any) for integration into the existing decoding solutions. The primary focus of our method is the short blocklength regime. In this setting, classical asymptotic constructions often fail to provide competitive performance, whereas our learned codes demonstrate a clear advantage over state-of-the-art solutions.

Beyond the conceptual novelty, we make three technical contributions: (i) we formulate the data-driven optimization objective adapted to the setting of interest (e.g., channel noise, code structure), (ii) we reformulate BP in a tensor fashion to learn the connectivity of the factor graph through backpropagation, and (iii) we propose a differentiable and fast optimization approach via a line-search method adapted to the relaxed binary programming setting. Applied to a wide variety of codes, our method produces codes that outperform existing codes on various channel noise settings, demonstrating the power and flexibility of the method in adapting to realistic settings of interest.

## 2 RELATED WORKS

Neural decoders typically focus on short and moderate-length codes for two main reasons. First, classical decoders reach the capacity of the channel for large codes, and second, the emergence of short data units applications driven by the Internet of Things (e.g., smart metering networks, messaging services, etc.) requires effective decoders for short to moderate-length codes. For example, 5G Polar codes have code lengths of 32 to 1024 (Liva et al., 2016; ESTI, 2021).

Previous work on neural decoders is generally divided into two main classes: model-free and model-based. Model-free decoders employ general types of neural network architectures (Cammerer et al., 2017; Gruber et al., 2017; Kim et al., 2018b; Bennatan et al., 2018; Jiang et al., 2019a; Choukroun & Wolf, 2022a). Model-based decoders implement parameterized versions of classical Belief Propagation (BP) decoders, where the Tanner graph is unfolded into an NN in which scalar weights are assigned to each variable edge. This results in an improvement in comparison to the baseline BP method for short codes (Nachmani et al., 2016; Raviv et al., 2020; Kwak et al., 2023). While model-based decoders benefit from a strong theoretical background, the architecture is overly restrictive. Also, the improvement gain generally vanishes for more iterations and longer codewords (Hoydis et al., 2022) and the integration cost remains very high due to both computational and memory requirements.

While neural decoders show improved performance in various communication settings, there has been very limited success in the design of novel neural coding methods, which remain impracticable for deployment (O'Shea & Hoydis, 2017; Kim et al., 2018a; Jiang et al., 2019b). Recently, (Choukroun & Wolf, 2024a) provided a new differentiable way of designing binary linear block codes (i.e., parity-check matrices) for a given neural decoder also showing improved performance with classical decoders.

Belief-propagation decoding has multiple advantages for LDPC codes (Gallager, 1962; Richardson & Urbanke, 2001). A large number of LDPC code (parity check matrix) design techniques exist in the literature, depending on the design criterion. Among them, Gallager (Gallager, 1962) de-

veloped the first regular LDPC codes as the concatenations of permuted sub-matrices. MacKay (MacKay & Neal, 1995) demonstrated the ability of sparse codes to reach near-capacity limits via semi-randomly generated matrices. Irregular LDPC codes have been developed by (Richardson et al., 2001; Luby et al., 2001; Chung et al., 2001) where the decoding threshold can be optimized via density-evolution. Progressive Edge Growth (Hu et al., 2001; 2005) has been proposed to design large girth codes. Certain classes of LDPC array codes have been presented in (Eleftheriou & Olcer, 2002) and LDPC codes with combinatorial design constraints have been developed in (Vasic & Milenkovic, 2004). Finite geometry codes have been developed in (Lucas et al., 2000; Kou et al., 2001) and repeat-accumulate codes have been proposed by Jin et al. (2000); Narayanaswami (2001). However, the classical methods are not data-driven and are difficult to adapt to the design of codes under constrained settings of interest (e.g., short codes, modern channels, structure constraints, etc.). Most related to our work are methods for structure learning (Koller & Friedman, 2009) for Bayesian networks such as the Chow-Liu Algorithm (Chow & Liu, 1968) or search-based methods (Tian, 2013). Related to greedy search-based methods, Elkelesh et al. (2019) recently suggested the application of classical genetic algorithms for the discovery of better IRA codes.

Density Evolution (DE) Richardson & Urbanke (2001) and EXIT Ten Brink (1999) methods estimate the expected performance of a code ensemble by tracking the evolution of statistical averages, such as the probability density functions of messages (DE) or the average mutual information (EXIT diagrams) as they pass through the iterative decoding process. A key feature is that these techniques are primarily designed for asymptotic analysis, assuming an infinite codeword length and an unlimited number of decoding iterations. Finally, the primary use is to numerically optimize macroscopic code parameters, such as the degree distributions, to maximize the iterative decoding threshold. Thus, DE/EXIT methods provide a theoretical upper bound on performance using approximations and asymptotic analysis, guiding the design of code ensembles.

Conversely, our ML-based approach directly optimizes finite-length, practical code designs or decoder parameters for a specific channel and a limited number of iterations by training on empirical data, aiming for the best practical performance rather than the asymptotic theoretical limit.

## 3 BACKGROUND

We assume a standard transmission protocol using a linear block code $C$. The code is defined by a generator matrix $G \in \{0, 1\}^{n \times k}$ and the parity check matrix $H \in \{0, 1\}^{(n-k) \times n}$ is defined such that $HG = 0$ over the order 2 Galois field $GF(2)$. The parity check matrix $H$ entails what is known as a Tanner graph (Tanner, 1981), which consists of $n$ variable nodes and $(n - k)$ check nodes. The edges of this bipartite graph correspond to the on-bits of the matrix $H$.

The input message $m \in \{0, 1\}^k$ (column vector) is encoded by $G$ to a codeword $c \in C \subset \{0, 1\}^n$ satisfying $Hc = H(Gm) = 0$ and transmitted via a Binary-Input Symmetric-Output channel, e.g., an AWGN channel. Let $y$ denote the channel output represented as $y = c_s + \varepsilon$, where $c_s$ denotes the transmission modulation of $c$ (e.g., Binary Phase Shift Keying (BPSK)), and $\varepsilon$ is random noise independent of the transmitted $c$. The main goal of the decoder $f_H : \mathbb{R}^n \to \mathbb{R}^n$ conditioned on the code (i.e., $H$) is to provide a soft approximation $\hat{x} = f_H(y)$ of the codeword.

The Belief Propagation algorithm allows the iterative transmission (propagation) of a current codeword estimate (belief) via a Trellis graph determined according to a *factor graph* defined by the code (i.e., the Tanner graph). The factor graph is unrolled into a Trellis graph, initiated with $n$ variable nodes, and composed of two types of interleaved layers defined by the check/factor nodes and variable nodes. An illustration of the Tanner graph unrolled to the Trellis graph is given in Figure 1.

As a message-passing algorithm, Belief Propagation operates on the Trellis graph by propagating the messages from variable nodes to check nodes and from check nodes to variable nodes, in an alternative and iterative fashion. The input layer generally corresponds to the vector of log-likelihood ratios (LLR) $L \in \mathbb{R}^n$ of the channel output $y$ defined as $L_v = \log \left( \Pr \left( c_v = 1 | y_v \right) / \Pr \left( c_v = 0 | y_v \right) \right)$.

Here, we describe ECC's classical notation of BP with, $v \in \{1, \ldots, n\}$ denotes the index corresponding to the $v^{th}$ element of the channel output $y$.

Let $x^i$ be the vector of messages that a column/layer in the Trellis graph propagates to the next one. At the first round of message passing, a variable node type of computation is performed such that

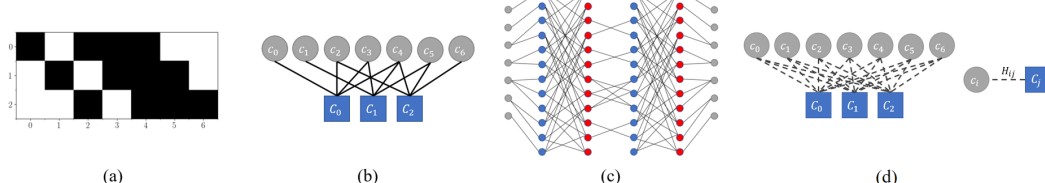

(a)             (b)             (c)             (d)

Figure 1: For the Hamming(7,4) Code: (a) parity check matrix, the induced (b) factor graph, and (c) the corresponding unrolled factor graph according to two rounds of message passing, with odd (even) layers in blue (red). In (d) our approach for structure learning via the learned binary weighting of the edges of the *complete* bipartite factor graph unlike the conventional sparse representation.

$$x_e^{2k+1} = x_{(c,v)}^{2k+1} = L_v + \sum_{e' \in N(v) \setminus \{(c,v)\}} x_{e'}^{2k}. \tag{1}$$

Here, each message indexed by the edge $e = (c, v)$ on the Tanner graph and $N(v) = \{(c, v) | H(c, v) = 1\}$, i.e, the set of all edges in which $v$ participates. By definition $x^0 = 0$ such that the messages are directly determined by the vector $L$ for $k = 0$.

For even layers, the check layer performs the following

$$x_e^{2k} = x_{(c,v)}^{2k} = 2\operatorname{arctanh}\left(\prod_{e' \in N(c) \setminus \{(c,v)\}} \tanh\left(\frac{x_{e'}^{2k-1}}{2}\right)\right) \tag{2}$$

where $N(c) = \{(c, v) | H(c, v) = 1\}$ is the set of edges in the Tanner graph in which row $c$ of the parity check matrix $H$ participates. The final $v^{th}$ output layer of the BP algorithm, which corresponds to the soft-decision output of the codeword, is given by $o_v = L_v + \sum_{e' \in N(v)} x_{e'}^{2L}$

## 4 METHOD

The performance of BP is strongly tied to the underlying Tanner graph induced by the code. BP and its variants are generally implemented over a *fixed* sparse graph, such that the only degree of freedom resides in the number of decoding iterations. While several recent contributions (Nachmani et al., 2016; Nachmani & Wolf, 2019) aim to enhance the BP algorithm by augmenting the Trellis graph with neural networks, these approaches assume and maintain fixed codes. Here, we propose optimizing the code for the BP algorithm on a decoding setting of interest. Given a trainable binary parity check matrix $H$, we wish to obtain BP-optimized codes by solving the following parameterized optimization problem

$$H^* = \underset{H \in \{0,1\}^{(n-k) \times n}}{\arg\min} \mathbb{E}_{m \sim \mathrm{Bern}^k(1/2), \varepsilon \sim \mathcal{Z}, T \in \mathbb{N}_+} \mathcal{D}\left(f_{H,T}\left(\phi(G(H)m) + \varepsilon\right), m\right) + \mathcal{R}(H), \tag{3}$$

Here, $G(H)$ denotes a generator matrix defined by $H$ (i.e., $HG^T = 0$), $\phi$ denotes the modulation function such that $c_s = \phi(c)$, and $\mathcal{Z}$ is the channel noise distribution. $f_{H,T}$ denotes the BP decoder built upon $H$ with $T$ iterations (sampled uniformly from a given target set), $\mathcal{D}$ denotes the distance metric of interest, and $\mathcal{R}$ denotes the potential hard/soft regularization of interest, e.g., sparsity or constraints on the code structure.

Several challenges arise from this optimization problem: (i) it is highly non-differentiable and results in an NP-hard binary non-linear integer programming problem, (ii) the codewords $c = Gm$ are both highly non-differentiable (matrix-vector multiplication over $GF(2)$ in case symmetry is not maintained during the optimization(Richardson & Urbanke, 2001)) and computationally expensive (inverse via Gaussian elimination of $H$), (iii) the modulation $\phi(\cdot)$ can be non-differentiable, and last but most important, (iv) BP assumes a *fixed* code (i.e., the factor graph edges) upon which the decoder is implemented. We note that the optimization relies on a distance measure applied within

the ECC pipeline and differs from Choukroun & Wolf (2024a) in the freedom of design of the parity matrix, the lack of modulation restriction, and the channel output processing.

**Learning the Factor graph via *Tensor* Belief *Back*propagation**    To obtain BP codes, we propose a *structure/Tanner graph learning* approach, where the bipartite graph is assumed as **complete** with **learnable** binary-weighted edges. This way, the tensor reformulation of BP weighted by $H$ allows a differentiable optimization of the Tanner graph itself. The two alternating stages of BP can now be represented in a differentiable matrix form rather than its static graph formulation, where the variable layers can be rewritten as

$$Q_{ij} = L_i + \sum_{j' \in C_i \setminus j} R_{j'i} \equiv L_i + \sum_{j'} R_{j'i} H_{j'i} - R_{ji}, \tag{4}$$

where $R_{ij}$ are the check layers, which are now represented as

$$R_{ji} = 2\mathrm{arctanh}\left(\prod_{i' \in V_j \setminus i} \tanh\left(\frac{Q_{i'j}}{2}\right)\right) = 2\mathrm{arctanh}\left(\frac{\prod_{i'}\left(\tanh\left(\frac{Q_{i'j}H_{ji'}}{2}\right) + (1 - H_{ji'})\right)}{\tanh\left(\frac{Q_{ij}}{2}\right)}\right), \tag{5}$$

where $C_i$ and $V_j$ correspond to the non-zero elements in column $i$ and row $j$ of $H$, respectively, while the ones elements in $(1 - H) \in \{0,1\}^{(n-k) \times n}$ satisfy the identity element of multiplication. Potential zero denominators have not been observed but can be handled via regularization or omission. As we can observe and assuming $H$ can be made differentiable, BP remains differentiable with respect to $H$ as a composition of differentiable functions. Algorithm 1 depicts the pseudo-code for the tensor formulation of the BP algorithm, implementing Eq. 5 and 4.

**Belief Propagation Codes Optimization**    The tensor reformulation solves the major challenge of graph learning (challenge (iv)). Challenges (ii) and (iii) are also eliminated in our formulation. First, since for any given $H$ the conditional independence of error probability under symmetry ((Richardson & Urbanke, 2001), Definition 1 and Lemma 1) is satisfied for message passing algorithms, it is enough to optimize the zero codeword only, i.e., $c = Gm = 0$, removing then any dependency on $G$ in the objective (challenge (ii)). As a byproduct, since the optimization is not dependent on the encoding or transmission stages, we obtain that the optimization problem is invariant to the choice of modulation, whether differentiable or not (challenge (iii)).

To optimize $H$ (challenge (i)) we relax the NP-hard binary programming problems to an unconstrained objective where, given a parameter matrix $\Omega \in \mathbb{R}^{(n-k) \times n}$, we have $H := H(\Omega) = \mathrm{bin}(\Omega)$. Here $\mathrm{bin}(\cdot)$ refers to the element-wise binarization operator implemented via the shifted straight-through-estimator (STE) (Bengio et al., 2013) defined such that $\mathrm{bin}(u) = (1 - \mathrm{sign}(u))/2$, with $\partial \mathrm{bin}(u)/\partial u := -0.5\mathbb{1}_{|u| \leq 1}$. Finally, opting for the binary cross-entropy loss (BCE) as a surrogate to the the Bit Error rate (BER) discrepancy measure $\mathcal{D} = \mathrm{BCE}$, we obtain the objective

$$\mathcal{L}(\Omega) = \sum_{t=1}^{T} \sum_{i=1}^{n} \mathrm{BCE}\left(f_{\mathrm{bin}(\Omega),t}\left(c_s + \varepsilon_i\right), c\right) + \mathcal{R}(\mathrm{bin}(\Omega)), \tag{6}$$

where $c_s = \phi(c)$ denotes the modulated *zero codeword* and $\varepsilon_i$ denotes the $i^{th}$ noise sample drawn from the channel noise distribution. This objective aims to provide optimal decoding on different numbers of (variable) decoding iterations $t$ (Nachmani et al., 2016). While highly non-convex, it is differentiable when considering the STE definition of the gradient (Bengio et al., 2013; Yin et al., 2019) and thus optimizable via classical first-order methods. Since $H$ is binary, only changes in the

---

**Algorithm 1:** Tensor Belief Propagation

```
1  function BP(llr, H, iters)
2      H = H.unsqueeze(dim=0).T
3      C = llr.unsqueeze(dim=-1)
4      for t in range(iters) do
5          Q = C if t == 0 else C + sum(R*H,dim=-1).unsqueeze(dim=-1) - R
6          tmp = tanh(0.5*Q)
7          R = 2*atanh( prod(tmp*H+(1-H),dim=1)/tmp )
8      return C.squeeze()+sum(R*H,dim=-1)
```

sign of $\Omega$ are relevant for the optimization, so most gradient descent iterations remain ineffective in reducing the objective using conventional small learning-rate regimes. Thus, given the gradient $\nabla_\Omega \mathcal{L}$, we propose a line-search procedure for finding the optimal step size.

**Binary Line-Search**    Conventional first-order optimization methods with small learning rate regimes have two major drawbacks with binarization (Rastegari et al., 2016; Courbariaux et al., 2016). First, they are generally slow since only gradient steps modifying the sign of the binarized tensor induce a modification of the loss. Second, they have difficulties in converging to local minima because of oscillating behavior around zero.

In general, efficient line search methods (Nocedal & Wright, 2006) assume local convexity or a smooth objective (Wolf, 1978) or, alternatively, apply exhaustive search on a given interval. Since this is not our case, we propose a novel efficient *grid*-search approach optimized to our binary programming setting. While classical grid search methods look for the optimal step size on handcrafted predefined sample points, in our binary setting we can search only for the step sizes *inducing a flip of the sign in* $\Omega$, provably limiting the maximum number of *relevant* grid samples to $n(n-k)$. Thus, the line-search problem is now given by

$$\lambda^* = \arg\min_{\lambda \in \mathcal{I}_\Omega} \mathcal{L}(\Omega - \lambda \nabla_\Omega \mathcal{L}), \quad \mathcal{I}_\Omega = \left\{ s_{ij} = \frac{(\Omega)_{ij}}{(\nabla_\Omega \mathcal{L})_{ij}} | s_{ij} > 0 \right\} \tag{7}$$

which corresponds to the (parallelizable) objective on the obtained discrete grid $\mathcal{I}_\Omega$. The same formulation can support other line-search objectives instead of the cross-entropy loss $\mathcal{L}$, such as the non-differentiable BER or Frame Error Rate (FER) instead of the Bayesian BCE loss. We provide in Appendix N a comparison of performance between SGD and the proposed line-search method.

**Training**    The optimization parameters are the following: the initial $H$ (i.e., initial $\Omega$), the maximum number of optimization steps (if convergence is not reached), the number and quality of the data samples, the grid search length, and the number of BP iterations.

We assume that an initial $H$ is given by the user as the code to be improved. The number of optimization steps is set to 20 iterations. The training noise is sampled randomly per batch in the $\{3, \dots, 7\}$ normalized SNR (i.e. $E_b/N_0$) range but can be modified according to the noise setting of interest. The number of data samples per optimization iteration is set to 4.9M for every code as sufficient gradient estimation, and the data samples are required to have non-zero syndrome. Because of computational constraints, the number of BP iterations during training is fixed and set to 5, while other ranges or values of interest can be used instead. For faster optimization, the grid search is heuristically restricted to the first 50 smallest step sizes as the optimal step size is generally in the vicinity of the working point (Appendix C) . Training and experiments are performed on $8 \times 12$GB GeForce RTX 2080 Ti GPUs and require 2.96 minutes on average per optimization step.

The full training algorithm (pseudocode) is given in Algorithm 2. Given an initial parity check matrix, the algorithm optimizes $H$ iteratively upon convergence. At each iteration, after computing the gradient on sufficiently large statistics (line 7), the line search procedure (line 10) searches for the optimal step size among those that flip the values of $H$ (line 9). A rank check of the parity matrix is applied to assert the matrix remains full-rank but it was never triggered.

---

**Algorithm 2:** Belief Propagation Codes Optimization

```
1  function Loss(H, x, y, BPiters=5)
2  │   return return BCE(BP(computeLLR(y), H, BPiters),x)

3  function BPCodesOptimization(H, iters)
4  │   Omega = 1-2*H
5  │   for t in range(iters) do
6  │   │   x,y = getData()
7  │   │   Loss(bin(Omega),x,y).backward()
8  │   │   lambdas = Omega/Omega.grad
9  │   │   lambdas = sorted(lambdas[lambdas>0].view(-1))[:50]
10 │   │   idx = argmin([Loss(bin(Omega -lambda*Omega.grad),x,y) for lambda in lambdas])
11 │   Omega = Omega - Omega.grad*lambdas[idx]
12 │   if converged:  break
13 │   return bin(Omega)
```

Table 1: A comparison of the negative natural logarithm of Bit Error Rate (BER) for several normalized SNR values of our method with classical codes. Higher is better. BP results are provided for 5 iterations in the first row and 15 in the second row. PEG$X$ means the degree of each node is of $X\%$ under the Progressive Edge Growth construction. The code format is Code($n$,$k$).

| Channel | AWGN | | | | | | Fading | | | | | | Bursting | | | | | |
|---|---|---|---|---|---|---|---|---|---|---|---|---|---|---|---|---|---|---|
| Method | BP | | | Our | | | BP | | | Our | | | BP | | | Our | | |
| $E_b/N_0$ | 4 | 5 | 6 | 4 | 5 | 6 | 4 | 5 | 6 | 4 | 5 | 6 | 4 | 5 | 6 | 4 | 5 | 6 |
| BCH(63,45) | 4.06 | 4.91 | 6.04 | **5.44** | **6.93** | **8.60** | 3.09 | 3.46 | 3.90 | **3.96** | **4.58** | **5.27** | 3.60 | 4.32 | 5.19 | **4.05** | **5.07** | **6.27** |
| | 4.21 | 5.24 | 6.59 | **5.70** | **7.35** | **9.16** | 3.13 | 3.55 | 4.04 | **4.10** | **4.80** | **5.56** | 3.67 | 4.52 | 5.59 | **4.21** | **5.40** | **6.85** |
| CCSDS(128,64) | 6.46 | 9.61 | 13.99 | **7.34** | **10.48** | **14.37** | 5.72 | 7.42 | 9.47 | **6.73** | **8.45** | **10.45** | 5.29 | 7.81 | 11.25 | **6.23** | **8.80** | **11.90** |
| | 7.32 | 10.83 | 15.43 | **8.61** | **12.26** | **16.00** | 6.43 | 8.29 | 10.28 | **8.05** | **10.07** | **12.37** | 5.98 | 8.85 | 12.53 | **7.39** | **10.43** | **13.28** |
| LDPC(121,60) | 4.81 | 7.17 | 10.75 | **7.70** | **10.87** | **14.25** | 4.10 | 5.23 | 6.68 | **6.68** | **8.47** | **10.50** | 3.97 | 5.75 | 8.40 | **6.23** | **8.89** | **11.98** |
| | 5.31 | 7.96 | 11.85 | **8.86** | **11.91** | **14.41** | 4.42 | 5.61 | 7.04 | **7.71** | **9.67** | **11.76** | 4.31 | 6.37 | 9.25 | **7.26** | **10.03** | **12.88** |
| LDPC(128,64) | 3.66 | 4.65 | 5.80 | **5.54** | **7.37** | **9.44** | 3.22 | 3.80 | 4.44 | **4.86** | **5.94** | **7.15** | 3.23 | 4.08 | 5.09 | **3.72** | **5.00** | **6.54** |
| | 4.00 | 5.16 | 6.42 | **6.56** | **8.70** | **10.81** | 3.51 | 4.18 | 4.84 | **5.64** | **6.85** | **8.14** | 3.48 | 4.51 | 5.66 | **4.13** | **5.72** | **7.66** |
| LDPC(96,48) | 6.73 | 9.48 | 12.98 | **7.22** | **9.96** | **13.37** | 3.83 | 4.57 | 5.35 | **5.37** | **6.51** | **7.71** | 5.68 | 7.94 | 10.90 | **5.90** | **8.19** | **10.91** |
| | 7.50 | 10.61 | **14.26** | **8.29** | **11.12** | 14.06 | 4.17 | 4.94 | 5.73 | **6.14** | **7.38** | **8.65** | 6.33 | 8.91 | **11.99** | **6.71** | **9.28** | 11.75 |
| LDPC PEG2(64,32) | 4.38 | 5.12 | 6.04 | **4.45** | **5.19** | **6.10** | 4.08 | 4.44 | 4.81 | **4.10** | **4.46** | **4.85** | 4.07 | 4.69 | 5.43 | **4.07** | **4.70** | **5.43** |
| | 4.38 | 5.13 | 6.04 | **4.44** | **5.19** | **6.10** | 4.08 | 4.44 | 4.81 | **4.10** | **4.47** | **4.85** | 4.06 | 4.69 | 5.43 | **4.06** | **4.69** | **5.44** |
| LDPC PEG5(64,32) | 6.02 | 8.20 | 10.95 | **6.53** | **8.73** | **11.56** | 5.63 | 6.86 | 8.31 | **6.22** | **7.48** | **8.82** | 5.18 | 6.97 | 9.34 | **5.59** | **7.41** | **9.51** |
| | 6.63 | 9.06 | **12.30** | **7.13** | **9.48** | 12.20 | 6.19 | 7.52 | 9.02 | **6.96** | **8.34** | **9.85** | 5.68 | 7.75 | **10.19** | **6.12** | **8.06** | 10.13 |
| LDPC PEG10(64,32) | 3.98 | 5.17 | 6.70 | **5.56** | **7.22** | **9.13** | 3.52 | 4.18 | 4.95 | **5.02** | **6.00** | **7.11** | 3.48 | 4.47 | 5.75 | **4.26** | **5.50** | **7.01** |
| | 4.27 | 5.77 | 7.67 | **6.25** | **8.28** | **10.59** | 3.71 | 4.47 | 5.30 | **5.60** | **6.72** | **7.90** | 3.67 | 4.90 | 6.46 | **4.73** | **6.22** | **8.01** |
| LTE(132,40) | 2.94 | 3.32 | 3.57 | **3.25** | **3.71** | **4.04** | 3.17 | 3.45 | 3.67 | **4.49** | **4.99** | **5.47** | 2.75 | 3.17 | 3.47 | **2.99** | **3.44** | **3.78** |
| | 3.37 | 3.79 | 4.09 | **3.93** | **4.49** | **4.89** | 3.60 | 3.82 | 4.01 | **5.32** | **5.81** | **6.31** | 3.17 | 3.62 | 3.96 | **3.53** | **4.03** | **4.41** |
| POLAR(128,86) | 3.76 | 4.17 | 4.58 | **4.83** | **5.87** | **6.58** | 3.15 | 3.53 | 3.91 | **3.64** | **4.28** | **4.94** | 3.48 | 3.96 | 4.37 | **3.69** | **4.51** | **5.18** |
| | 4.02 | 4.67 | 5.38 | **5.37** | **6.88** | **8.10** | 3.28 | 3.73 | 4.18 | **3.92** | **4.70** | **5.52** | 3.65 | 4.31 | 4.97 | **3.87** | **4.91** | **5.91** |
| RS(60,52) | 4.41 | 5.32 | 6.41 | **5.02** | **6.38** | **7.99** | 3.11 | 3.41 | 3.77 | **3.37** | **3.73** | **4.12** | 3.85 | 4.58 | 5.44 | **4.17** | **5.18** | **6.40** |
| | 4.54 | 5.52 | 6.64 | **5.07** | **6.47** | **8.12** | 3.13 | 3.43 | 3.81 | **3.38** | **3.75** | **4.15** | 3.91 | 4.72 | 5.67 | **4.21** | **5.27** | **6.56** |

## 5 EXPERIMENTS

Our framework is evaluated on five classes of linear codes: various Low-Density Parity Check (LDPC) codes (Gallager, 1962; Abu-Surra et al., 2010), Polar codes (Arikan, 2008), Reed Solomon codes (Reed & Solomon, 1960), Bose–Chaudhuri–Hocquenghem (BCH) codes (Bose & Ray-Chaudhuri, 1960), LTE Turbo codes and random codes. All the parity check matrices are taken from (Helmling et al., 2019) except the LDPC codes created using the popular Progressive Edge Growth framework (Hu et al., 2005; MacKay). Code is available in supplementary materials.

We consider three types of channel noise under BPSK modulation. (i) The canonical AWGN channel given as $y = c_s + \varepsilon$ with $\varepsilon \sim \mathcal{N}(0, \sigma I_n)$. (ii) The Rayleigh fading channel, where $y = h \odot c_s + \varepsilon$, with $h$ the iid Rayleigh distributed fading vector with coefficient 1 and $\varepsilon$ the regular AWGN noise, where we assume ideal channel state information. Finally, (iii) AWGN channel with Gaussian mixture channel (also referred to as bursty noise (Kurmukova & Gunduz, 2024)) simulating wireless channel interference as $y = c_s + \varepsilon + \zeta$ with $\varepsilon$ the AWGN and $\zeta_i \sim \mathcal{N}(0, \sqrt{2}\sigma)$ with probability $\rho = 0.1$ and $\zeta_i = 0$ with probability $1 - \rho$.

The results are reported as negative natural logarithm bit error rates (BER) for three different normalized SNR values ($E_b/N_0$), following the conventional testing benchmark, e.g., (Nachmani & Wolf, 2019; Choukroun & Wolf, 2022a). BP-based results are obtained after $\ell = 5$ BP iterations in the first row and $\ell = 15$ in the second row of the results tables. During testing, at least $10^5$ random codewords are decoded, to obtain at least 50 frames with errors at each SNR value. For this section, we performed a small hyperparameter search as reported in Appendix A, where the final code is selected to have the lowest average BER on the SNR test range.

Table 1 lists the results. *Our* method means BP applied to the learned code initialized by the given classical code. Evidently, our method improves by large margins all code families on the three different channel noise scenarios and with both numbers of decoding iterations, demonstrating the capacity of the framework to provide improved codes on multiple settings of interest. Appendix B has the same table with a broader SNR range ($E_b/N_0 \in \{3, \ldots, 7\}$). Appendix I presents the standard visualization of the BER and BLER vs $E_b/N_0$ curves on several codes. Performance on larger number of decoding iterations is given in Appendix K. The overall improvement statistics (including

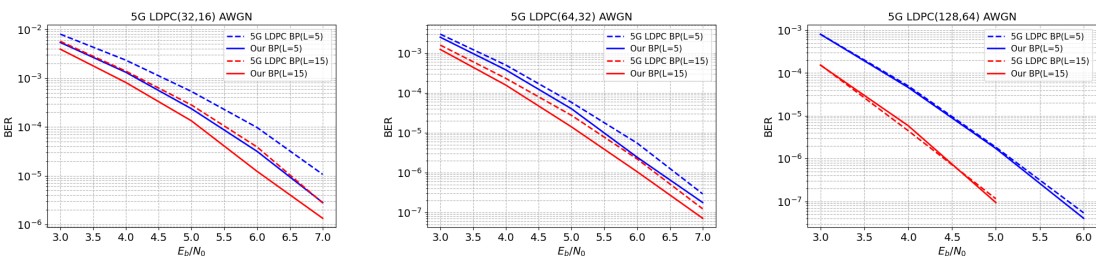

Figure 2: BER and BLER performance of the method on different 5G LDPC codes.

SD) in dB on all the sparse codes and all the codes is in Appendix F. For completeness, Appendix H evaluates the genetic algorithm of Elkelesh et al. (2019); our method demonstrates much better performance while being faster by orders of magnitude. Appendix M shows the applicability of learned codes to different channels. Finally, we observed that generally the learned codes exhibit similar error-floor behavior (the training range remains outside of the error-floor range, though).

We further show the performance of the proposed method on the state-of-the-art 5G NR LDPC short blocklenght codes (Richardson & Kudekar, 2018; 3GPP, 2018) in Figure 2. Our method optimizes codes via first-order gradient descent, within highly non-convex loss landscapes. As a result, with well-initialized LDPC codes (e.g., longer than 128, structured ones), the optimization often remains/converges to a nearby local minimum.

Table 2 shows that our method can outperform the performance of the SCL algorithm (Tal & Vardy, 2015) even on very short codes where the performances are close to ML decoding. The SCL results are obtained using the implementation of Cassagne et al. (2019). We provide the performance of BP and of our method with the same Polar code initialization (BP (Polar) and Our(Polar)) and with 5G LDPC code initialization (BP (5G LDPC), Our (5G LDPC)). CRC augmentation is not used to ensure a fair comparison with our LDPC and optimized codes, which also do not incorporate CRC or other augmentation techniques (e.g., automorphism). SCL performance is provided with the corresponding Polar code.

Evidently, even in the extremely short length setting where sparsity is hard to obtain our method is able to remarkably improve the performance over existing short-length low-density codes and get close to the ML bound even within very few iterations, even with bad initialization. With good initialization (good sparse code), our method provides state-of-the-art performance.

Table 2: A comparison of the negative natural logarithm of BER for several normalized SNR values of our method with classical codes and SCL decoding. The first and the second row of the SCL algorithm denote performance with a list length of 1 and 32 respectively.

| Method | SCL | | | BP (Polar) | | | Our (Polar) | | | BP (5G LDPC) | | | Our (5G LDPC) | | |
|---|---|---|---|---|---|---|---|---|---|---|---|---|---|---|---|
| $E_b/N_0$ | 4 | 5 | 6 | 4 | 5 | 6 | 4 | 5 | 6 | 4 | 5 | 6 | 4 | 5 | 6 |
| (32,16) | 6.22 | 8.06 | 10.28 | 4.36 | 5.59 | 7.18 | 5.48 | 7.02 | 8.92 | 6.06 | 7.53 | 9.23 | 6.63 | 8.53 | 10.36 |
| | 7.74 | 10.41 | 13.34 | 4.64 | 6.07 | 7.94 | 5.76 | 7.44 | 9.41 | 6.57 | 8.17 | 10.16 | 7.11 | 8.92 | 11.31 |
| (64,32) | 7.36 | 9.82 | 12.98 | - | - | - | - | - | - | 7.59 | 9.75 | 12.10 | 7.87 | 10.12 | 12.92 |
| | 8.10 | 10.73 | 14.00 | - | - | - | - | - | - | 8.36 | 10.50 | 13.02 | 8.75 | 11.17 | 13.76 |
| (128,64) | 8.49 | 11.46 | 16.16 | - | - | - | - | - | - | 9.90 | 13.20 | 16.73 | 9.98 | 13.27 | 17.02 |
| | 9.59 | 13.12 | 17.48 | - | - | - | - | - | - | 12.31 | 15.98 | 18.06 | 12.04 | 16.18 | 18.66 |

## 6 ANALYSIS

**Initialization and Random Codes**     Figure 3 presents performance on random codes initialized with different sparsity rates. The parity check matrix is initialized in a systematic form $H = [I_{n-k}, P]$ for full rank initialization, where $P \sim \text{Bern}^{(n-k) \times k}(p)$. Evidently, the framework can greatly improve the performance of the original random code. Most importantly, different initializations provide convergence to different local optima, and better initialization generally induces

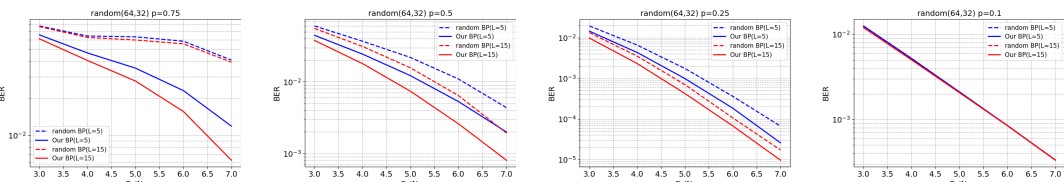

Figure 3: Performance of the method on random codes under different sparsity rate initialization $p$.

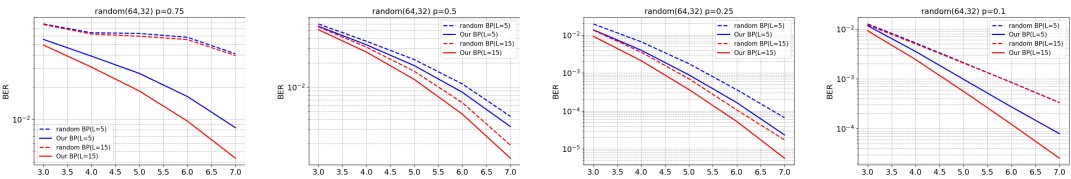

Figure 4: Constrained systematic random codes under different sparsity rate initialization $p$.

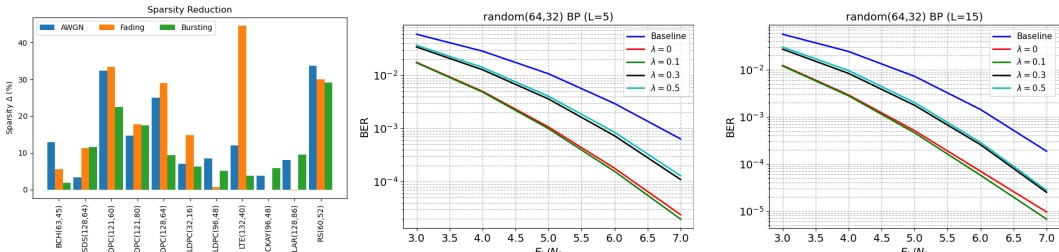

Figure 5: Sparsity reduction of the proposed codes.

Figure 6: Effect of different values of the $L_1$ regularization factor $\lambda$ for random codes with $p = 0.25$ on the AWGN channel.

convergence to a better minimum. Performance on other code lengths is provided in Appendix G. Finally, as a limitation of first order methods, good initialization (i.e., (large) well-performing sparse codes under BP decoding) requires perturbation at initialization or during training (c.f., Appendix A) to escape local minima (escaping local minima is an open challenge for deep learning in general).

**Constrained Codes** Figure 4 depicts the performance of the method on constrained systematic random codes as described in the previous paragraph, while here, we constrain them to maintain their systematic form during the optimization, i.e., only the parity matrix elements of $P$ are optimized (via $\Omega$). The optimization is performed by backpropagating over the $P$ tensor only, similarly to having a hard structure constraint on the identity part of $H$. While maintaining a structure of interest, we can observe this regularization can further improve the convergence quality (e.g., $p = 0.1$) compared to the unconstrained setting of Figure 3. Performance on other code lengths is provided in Appendix G. Any constraint (e.g., dual diagonal) can be similarly added to the code.

Figure 5 presents the sparsification of the codes created by the framework. Here, $\Delta = 100(S_b - S_o)/S_b$ represents the sparsity ratio, with $S_b$ and $S_o$ being the sparsity of the baseline code and our code, respectively. As can be seen, optimization always provides sparser codes. Nevertheless, the optimization does not modify the girth of the code. Appendix F lists the column weights distribution of the initial and learned parity check matrices.

Figure 6 presents the performance of the method on random codes with sparsity constraint, i.e., $\mathcal{R}(H) = \lambda \|H\|_1$, with $\lambda \in \mathbb{R}_+$. Evidently, adding a sparsity constraint is generally not profitable since the optimization over BP already induces sparse codes, suggesting that gradient descent's inductive bias and BP have similar sparsity enforcement effects and performance.

**Learned Codes Visualization** Appendix L depicts the learned codes. For low-density codes, the modifications remain small, since the code is near a local optimum, while for denser codes, the change can be substantial. Additionally, the optimized codes tend to be sparser than the original.

Appendix C depicts the **line search optimization**, demonstrating the high non-convexity and the proximity of the optimum to the current estimate. Appendix Dhas statistics on **convergence rates** demonstrating fast monotonic convergence. Finally, Appendix E lists the performance of the learned

code on the efficient **Min-Sum approximation** of the BP algorithm and shows that the learned code outperforms the baseline codes over the Min-Sum framework as well.

# 7 CONCLUSIONS

We present a novel gradient-based optimization method of binary linear block codes for the Belief Propagation algorithm. The proposed framework enables the differentiable optimization of the factor graph via weighted tensor representation. The optimization is efficiently carried out via a tailor-made grid search procedure that is aware of the binary constraint of the optimization problem.

A common criticism of ML-based ECC is that the neural decoder cannot be deployed directly without the application of massive deep-learning acceleration methods. Here, we show that the code can be designed efficiently in a data-driven fashion on differentiable formulations of classical decoders. The optimization of codes may open the door to the establishment of new industry standards and the creation of new families of codes. Future work includes the development of more efficient optimization methods, able to define better initializations and to escape bad local minima.

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

## A  HYPER-PARAMETER TUNING

Under the problem's stochastic optimization, we provide here the different modifications used to obtain better performance. The first set of training/optimization hyperparameters is the $E_b/N_0$ range defined as $(u, 7)$ with $u \in \{3, 4, 5\}$. The second set of hyperparameters is the data sampling, where we experimented with random data (i.e., classical setting) and data with non-zero syndromes only. Finally, for better backpropagation, we also experimented with a soft approximation $\tilde{H}$ of the binary $H$ during the optimization, defined as

$$\tilde{H}_{ij} = \begin{cases} (-1)^z \epsilon, & if \quad H_{ij} = 0 \\ 1, & else \end{cases}$$

where $z \sim Bern(0.5)$ and $\epsilon$ is a small scalar ($10^{-7}$ in our experiments). We note we only used a size 15 random subset of all the possible permutations of the hyperparameters mentioned above.

## B  MORE SNR RESULTS

We provide results on a larger range of SNRs in Table 3.

Table 3: A comparison of the negative natural logarithm of Bit Error Rate (BER) for several normalized SNR values of our method with classical codes. Higher is better. BP results are provided for 5 iterations in the first row and 15 in the second row. The best results are in bold.

| Channel / Method / $E_b/N_0$ | AWGN BP 3 | 4 | 5 | 6 | 7 | AWGN Our 3 | 4 | 5 | 6 | 7 | Fading BP 3 | 4 | 5 | 6 | 7 | Fading Our 3 | 4 | 5 | 6 | 7 | Bursting BP 3 | 4 | 5 | 6 | 7 | Bursting Our 3 | 4 | 5 | 6 | 7 |
|---|---|---|---|---|---|---|---|---|---|---|---|---|---|---|---|---|---|---|---|---|---|---|---|---|---|---|---|---|---|---|
| BCH(63,45) | 3.35 | 4.06 | 4.91 | 6.04 | 7.47 | **4.23** | **5.44** | **6.93** | **8.60** | **10.27** | 2.77 | 3.09 | 3.46 | 3.90 | 4.37 | **3.42** | **3.96** | **4.58** | **5.27** | **5.99** | 3.00 | 3.60 | 4.32 | 5.19 | 6.25 | **3.24** | **4.05** | **5.07** | **6.27** | **7.67** |
|  | 3.40 | 4.21 | 5.24 | 6.59 | 8.35 | **4.36** | **5.70** | **7.35** | **9.16** | **11.10** | 2.79 | 3.13 | 3.55 | 4.04 | 4.61 | **3.50** | **4.10** | **4.80** | **5.56** | **6.36** | 3.02 | 3.67 | 4.52 | 5.59 | 6.93 | **3.31** | **4.21** | **5.40** | **6.85** | **8.55** |
| CCSDS(128,64) | 4.32 | 6.46 | 9.61 | 13.99 | **18.27** | **4.99** | **7.34** | **10.48** | **14.37** | 17.38 | 4.37 | 5.72 | 7.42 | 9.47 | 11.84 | **5.22** | **6.73** | **8.45** | **10.45** | **12.36** | 3.62 | 5.29 | 7.81 | 11.25 | **15.59** | **4.32** | **6.23** | **8.80** | **11.90** | 14.76 |
|  | 4.82 | 7.32 | 10.83 | 15.43 | **18.51** | **5.80** | **8.61** | **12.26** | **16.00** | 18.15 | 4.89 | 6.43 | 8.29 | 10.28 | 12.88 | **6.22** | **8.05** | **10.07** | **12.37** | **14.87** | 3.97 | 5.98 | 8.85 | 12.53 | **17.10** | **5.05** | **7.39** | **10.43** | **13.28** | 15.56 |
| LDPC(121,60) | 3.33 | 4.81 | 7.17 | 10.75 | 15.69 | **5.29** | **7.70** | **10.87** | **14.25** | **16.82** | 3.24 | 4.10 | 5.23 | 6.68 | 8.56 | **5.18** | **6.68** | **8.47** | **10.50** | **12.31** | 2.87 | 3.97 | 5.75 | 8.40 | 12.16 | **4.32** | **6.23** | **8.89** | **11.98** | **14.91** |
|  | 3.53 | 5.31 | 7.96 | 11.85 | 17.01 | **6.18** | **8.86** | **11.91** | **14.41** | **17.04** | 3.44 | 4.42 | 5.61 | 7.04 | 8.77 | **6.04** | **7.71** | **9.67** | **11.76** | **14.21** | 2.98 | 4.31 | 6.37 | 9.25 | 13.10 | **5.02** | **7.26** | **10.03** | **12.88** | **15.14** |
| LDPC(121,80) | 4.50 | 6.59 | 9.68 | 13.43 | **18.51** | **5.28** | **7.77** | **11.21** | **15.06** | 18.40 | 3.67 | 4.60 | 5.80 | 7.22 | 8.95 | **4.42** | **5.55** | **6.90** | **8.36** | **10.00** | 3.74 | 5.30 | 7.60 | 10.66 | 14.88 | **4.32** | **6.23** | **8.87** | **12.19** | **16.31** |
|  | 4.85 | 7.35 | 10.94 | 15.46 | **19.61** | **5.85** | **8.75** | **12.45** | **15.67** | 18.30 | 3.89 | 4.97 | 6.29 | 7.82 | 9.58 | **4.89** | **6.25** | **7.80** | **9.47** | **11.21** | 3.94 | 5.81 | 8.50 | 12.15 | 16.45 | **4.73** | **6.99** | **10.09** | **13.74** | **17.55** |
| LDPC(128,64) | 2.88 | 3.66 | 4.65 | 5.80 | 7.03 | **4.07** | **5.54** | **7.37** | **9.44** | **11.71** | 2.73 | 3.22 | 3.80 | 4.44 | 5.14 | **3.95** | **4.86** | **5.94** | **7.15** | **8.38** | 2.58 | 3.23 | 4.08 | 5.09 | 6.21 | **2.80** | **3.72** | **5.00** | **6.54** | **8.30** |
|  | 3.04 | 4.00 | 5.16 | 6.42 | 7.77 | **4.71** | **6.56** | **8.70** | **10.81** | **12.92** | 2.90 | 3.51 | 4.18 | 4.84 | 5.54 | **4.55** | **5.64** | **6.85** | **8.14** | **9.59** | 2.68 | 3.48 | 4.51 | 5.66 | 6.88 | **2.97** | **4.13** | **5.72** | **7.66** | **9.86** |
| LDPC(32,16) | 3.45 | 4.36 | 5.59 | 7.18 | 9.19 | **4.29** | **5.48** | **7.02** | **8.92** | **11.23** | 3.44 | 4.03 | 4.70 | 5.47 | 6.31 | **4.53** | **5.26** | **6.02** | **6.82** | **7.61** | 3.10 | 3.88 | 4.89 | 6.18 | 7.82 | **3.78** | **4.77** | **6.02** | **7.52** | **9.22** |
|  | 3.59 | 4.64 | 6.07 | 7.94 | 10.23 | **4.47** | **5.76** | **7.44** | **9.41** | **12.03** | 3.62 | 4.29 | 5.06 | 5.90 | 6.83 | **4.67** | **5.43** | **6.23** | **6.97** | **7.81** | 3.21 | 4.09 | 5.26 | 6.76 | 8.58 | **3.93** | **5.01** | **6.35** | **7.96** | **9.72** |
| LDPC(96,48) | 4.72 | 6.73 | 9.48 | 12.98 | **16.87** | **5.17** | **7.22** | **9.96** | **13.37** | 16.45 | 3.19 | 3.83 | 4.57 | 5.35 | 6.17 | **4.38** | **5.37** | **6.51** | **7.71** | **8.94** | 4.03 | 5.68 | 7.94 | 10.90 | **14.27** | **4.23** | **5.90** | **8.19** | **10.91** | 13.61 |
|  | 5.20 | 7.50 | 10.61 | **14.26** | **17.80** | **5.85** | **8.29** | **11.12** | 14.06 | 17.19 | 3.44 | 4.17 | 4.94 | 5.73 | 6.58 | **4.99** | **6.14** | **7.38** | **8.65** | **9.77** | 4.40 | 6.33 | 8.91 | **11.99** | **15.55** | **4.71** | **6.71** | **9.28** | 11.75 | 14.06 |
| LTE(132,40) | 2.49 | 2.94 | 3.32 | 3.57 | 3.81 | **2.72** | **3.25** | **3.71** | **4.04** | **4.36** | 2.82 | 3.17 | 3.45 | 3.67 | 3.89 | **3.97** | **4.49** | **4.99** | **5.47** | **5.96** |  |  |  |  |  | **2.48** | **2.99** | **3.44** | **3.78** | **4.05** |
|  | 2.85 | 3.37 | 3.79 | 4.09 | 4.32 | **3.26** | **3.93** | **4.49** | **4.89** | **5.22** | 3.29 | 3.60 | 3.82 | 4.01 | 4.21 | **4.78** | **5.32** | **5.81** | **6.31** | **6.84** | 2.63 | 3.17 | 3.62 | 3.96 | 4.19 | **2.92** | **3.53** | **4.03** | **4.41** | **4.70** |
| MACKAY(96,48) | 4.77 | 6.75 | 9.45 | **12.85** | **16.37** | **5.03** | **7.03** | **9.63** | 12.78 | 16.11 | 4.98 | 6.28 | 7.86 | 9.55 | 11.30 | **5.18** | **6.53** | **8.06** | **9.77** | **11.58** | 4.08 | 5.72 | 7.97 | 10.81 | 13.92 | **4.28** | **5.95** | **8.23** | **10.91** | **14.02** |
|  | 5.28 | 7.59 | 10.52 | **14.09** | 17.43 | **5.63** | **7.99** | **10.97** | 14.05 | **17.49** | 5.55 | 7.04 | 8.76 | 10.64 | 12.58 | **5.93** | **7.47** | **9.32** | **11.19** | **13.14** | 4.47 | 6.39 | 8.90 | 11.91 | 15.23 | **4.79** | **6.82** | **9.41** | **12.71** | **15.81** |
| POLAR(128,86) | 3.25 | 3.76 | 4.17 | 4.58 | 5.12 | **3.72** | **4.83** | **5.87** | **6.58** | **7.21** | 2.80 | 3.15 | 3.53 | 3.91 | 4.26 | **3.10** | **3.64** | **4.28** | **4.94** | **5.58** | 2.95 | 3.48 | 3.96 | 4.37 | 4.78 | **2.96** | **3.69** | **4.51** | **5.18** | **5.73** |
|  | 3.36 | 4.02 | 4.67 | 5.38 | 6.19 | **3.96** | **5.37** | **6.88** | **8.10** | **9.00** | 2.87 | 3.28 | 3.73 | 4.18 | 4.60 | **3.26** | **3.92** | **4.70** | **5.52** | **6.30** | 3.03 | 3.65 | 4.31 | 4.97 | 5.66 | **3.03** | **3.87** | **4.91** | **5.91** | **6.88** |
| RS(60,52) | 3.65 | 4.41 | 5.32 | 6.41 | 7.80 | **3.98** | **5.02** | **6.38** | **7.99** | **9.73** | 2.86 | 3.11 | 3.41 | 3.77 | 4.16 | **3.05** | **3.37** | **3.73** | **4.12** | **4.53** | 3.26 | 3.85 | 4.58 | 5.44 | 6.43 | **3.42** | **4.17** | **5.18** | **6.40** | **7.75** |
|  | 3.70 | 4.54 | 5.52 | 6.64 | 8.04 | **4.00** | **5.07** | **6.47** | **8.12** | **9.80** | 2.87 | 3.13 | 3.43 | 3.81 | 4.24 | **3.06** | **3.38** | **3.75** | **4.15** | **4.56** | 3.27 | 3.91 | 4.72 | 5.67 | 6.78 | **3.42** | **4.21** | **5.27** | **6.56** | **8.01** |
| PGE2(64,32) | 3.78 | 4.38 | 5.12 | 6.04 | 7.17 | **3.84** | **4.45** | **5.19** | **6.10** | **7.22** | 3.74 | 4.08 | 4.44 | 4.81 | 5.20 | **3.75** | **4.10** | **4.46** | **4.85** | **5.24** | 3.54 | 4.07 | 4.69 | 5.43 | 6.29 | **3.54** | **4.07** | **4.70** | **5.43** | **6.30** |
|  | 3.78 | 4.38 | 5.13 | 6.04 | 7.16 | **3.84** | **4.44** | **5.19** | **6.10** | **7.23** | 3.74 | 4.08 | 4.44 | 4.81 | 5.20 | **3.75** | **4.10** | **4.47** | **4.85** | **5.24** | 3.54 | 4.06 | 4.69 | 5.43 | 6.29 | **3.54** | **4.06** | **4.69** | **5.44** | **6.30** |
| PGE5(64,32) | 4.41 | 6.02 | 8.20 | 10.95 | 14.40 | **4.82** | **6.53** | **8.73** | **11.56** | **14.41** | 4.56 | 5.63 | 6.86 | 8.31 | 9.78 | **5.09** | **6.22** | **7.48** | **8.82** | **10.16** | 3.84 | 5.18 | 6.97 | 9.34 | **12.25** | **4.19** | **5.59** | **7.41** | **9.51** | 11.84 |
|  | 4.78 | 6.63 | 9.06 | **12.30** | **15.75** | **5.26** | **7.13** | **9.48** | 12.20 | 14.90 | 4.98 | 6.19 | 7.52 | 9.02 | 10.76 | **5.67** | **6.96** | **8.34** | **9.85** | **11.24** | 4.13 | 5.68 | 7.75 | **10.19** | **13.40** | **4.55** | **6.12** | **8.06** | 10.13 | 12.27 |
| PGE10(64,32) | 3.08 | 3.98 | 5.17 | 6.70 | 8.49 | **4.21** | **5.56** | **7.22** | **9.13** | **11.11** | 2.96 | 3.52 | 4.18 | 4.95 | 5.81 | **4.16** | **5.02** | **6.00** | **7.11** | **8.33** | 2.75 | 3.48 | 4.47 | 5.75 | 7.28 | **3.30** | **4.26** | **5.50** | **7.01** | **8.80** |
|  | 3.20 | 4.27 | 5.77 | 7.67 | 9.72 | **4.62** | **6.25** | **8.28** | **10.59** | **12.84** | 3.08 | 3.71 | 4.47 | 5.30 | 6.24 | **4.60** | **5.60** | **6.72** | **7.90** | **9.24** | 2.82 | 3.67 | 4.90 | 6.46 | 8.26 | **3.57** | **4.73** | **6.22** | **8.01** | **10.06** |

## C  LINE SEARCH OPTIMIZATION

In Figure 7 we provide visualizations of the line search procedure. We provide BER with respect to the step size $\lambda_i$ indexed by $i$ ($\lambda_0 \equiv 0$). We can observe the high non-convexity of the objective, with the presence of several local minima. We can also notice the proximity of the optimum to the current parity-check estimate (i.e., $\lambda_0$).

## D  CONVERGENCE RATE

In Figure 8 we provide statistics on the number of optimization iterations for convergence (a). We also provide (b,c,d) typical convergence. We can observe that the framework typically converges within a few iterations and that the loss decreases monotonically.

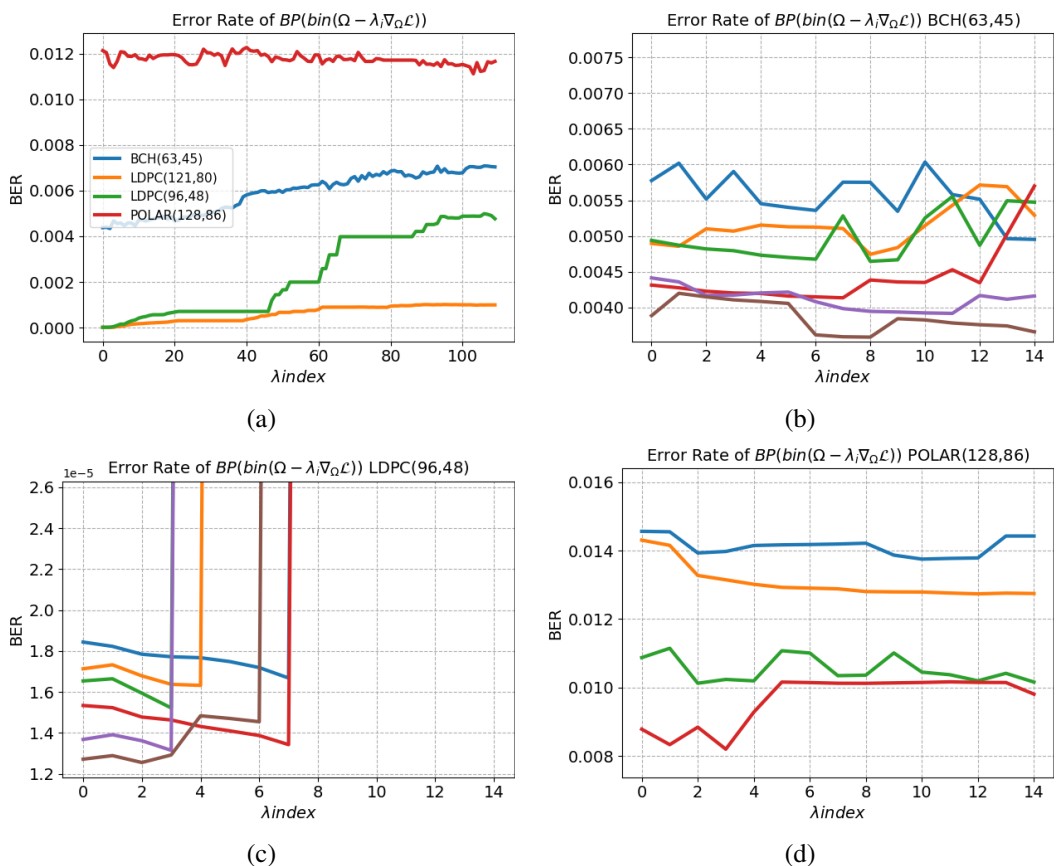

Figure 7: BER in function of the step size index $i$ on AWGN channel. (a) Averaged BER over the optimization iterations for 4 codes. (b,c,d) BER per optimization iteration for the first 5 optimization iterations and the first 10 indices for three different codes. Here $\lambda_0 = 0$ denotes the original BER.

Table 4: A comparison of the negative natural logarithm of Bit Error Rate (BER) for five normalized SNR values of our method applied on the Min-Sum BP algorithm. $NE$ = no errors spotted under the testing limits.

| BP Method | Sum-Product | | | | | | | | | | Min-Sum | | | | | | | | | |
|---|---|---|---|---|---|---|---|---|---|---|---|---|---|---|---|---|---|---|---|---|
| Method | Baseline | | | | | Our | | | | | Baseline | | | | | Our | | | | |
| $E_b/N_0$ | 3 | 4 | 5 | 6 | 7 | 3 | 4 | 5 | 6 | 7 | 3 | 4 | 5 | 6 | 7 | 3 | 4 | 5 | 6 | 7 |
| BCH(63,45) | 3.35 | 4.06 | 4.92 | 5.98 | 7.39 | 3.48 | 4.30 | 5.29 | 6.51 | 8.12 | 3.04 | 3.79 | 4.89 | 6.33 | 8.13 | 3.21 | 4.09 | 5.32 | 6.84 | 8.65 |
| | 3.40 | 4.22 | 5.24 | 6.60 | 8.33 | 3.57 | 4.49 | 5.69 | 7.17 | 9.17 | 3.22 | 4.09 | 5.41 | 7.06 | 9.14 | 3.40 | 4.44 | 5.89 | 7.60 | 10.00 |
| CCSDS(128,64) | 4.32 | 6.47 | 9.62 | 13.80 | 18.40 | 4.44 | 6.66 | 9.73 | 13.60 | 18.30 | 4.21 | 6.62 | 10.40 | 15.10 | 19.40 | 4.35 | 6.82 | 10.50 | 15.00 | 21.00 |
| | 4.82 | 7.30 | 10.70 | 15.50 | 17.90 | 4.99 | 7.57 | 11.00 | 15.60 | $NE$ | 4.76 | 7.66 | 12.20 | 17.70 | $NE$ | 4.97 | 8.03 | 12.30 | 17.40 | $NE$ |
| LDPC(32,16) | 3.46 | 4.39 | 5.60 | 7.20 | 9.23 | 3.62 | 4.59 | 5.83 | 7.45 | 9.52 | 3.36 | 4.38 | 5.75 | 7.65 | 10.10 | 3.53 | 4.61 | 6.01 | 7.92 | 10.10 |
| | 3.61 | 4.66 | 6.07 | 7.87 | 10.30 | 3.80 | 4.91 | 6.36 | 8.16 | 10.70 | 3.55 | 4.69 | 6.21 | 8.21 | 10.90 | 3.74 | 4.93 | 6.50 | 8.44 | 11.10 |
| LDPC(96,48) | 4.70 | 6.73 | 9.52 | 13.20 | 17.30 | 5.01 | 7.11 | 9.92 | 13.50 | 16.90 | 4.71 | 6.96 | 9.95 | 14.20 | 18.30 | 4.98 | 7.16 | 10.10 | 14.00 | 15.80 |
| | 5.20 | 7.55 | 10.70 | 14.40 | 18.50 | 5.70 | 8.13 | 11.30 | 14.70 | 17.40 | 5.23 | 7.89 | 11.50 | 15.10 | 19.10 | 5.68 | 8.32 | 12.00 | 14.80 | 15.80 |

## E  IMPACT ON OTHER BP VARIANTS

In Table 4 we provide the performance of the learned code on the more efficient Min-Sum approximation of the Sum-Product algorithm. We can observe that the codes learned with BP consistently outperform the performance of the Min-Sum approximation as well. For some codes, the training range may need to be adjusted. We note our method can be applied to neural BP decoders as well. The direct optimization over BP approximations and augmentations is left for future work.

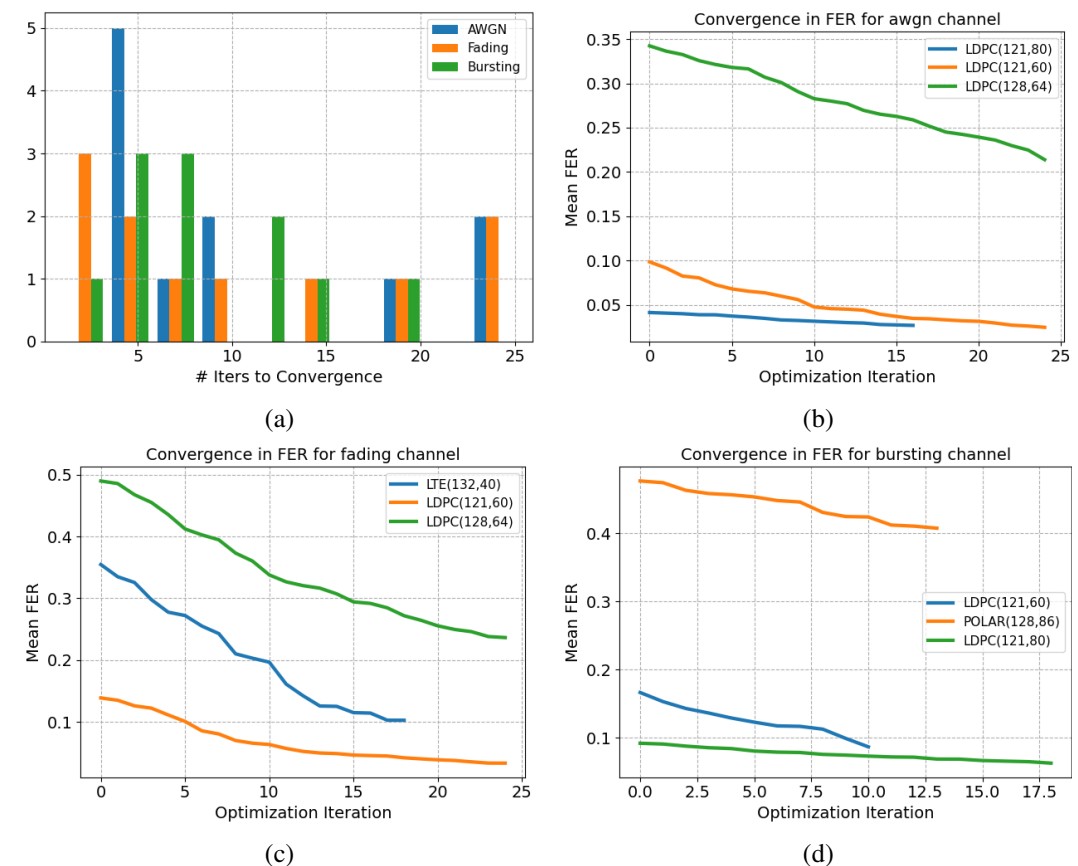

Figure 8: (a) Histogram of the number of required iterations until convergence. (b) Convergence rate of the Frame Error Rate for three codes on (b) AWGN, (c) fading, and (d) bursting channel. We selected the three codes with the largest number of iterations. The FER is averaged over all the tested $E_b/N_0 = \{3, \ldots 7\}$ range.

## F  IMPROVEMENT STATISTICS ON ALL THE CODES

We provide in Figure 10 the statistics of improvement on sparse codes in 9 and all the codes presented in Table 3.

## G  MORE RANDOM CODES

We provide in Figure 11 the performance of the proposed method on random codes initialized with different sparsity rates on different lengths. We also provide in Figure 12 the performance of the proposed method on constrained systematic random codes initialized with different sparsity rates on different lengths.

## H  COMPARISON WITH GENETIC ALGORITHM

We provide in table 5 a comparison with the genetic algorithm of Elkelesh et al. (2019). We note that the method requires 230 offspring/code evaluations per iteration, with 300 iterations (Fig. 7 in (Elkelesh et al., 2019)) or even an infinite loop (cf. the provided MATLAB code). Our algorithm is tested on 50 line-search steps as described in in the paper on 2 to 25 iterations (cf. App. D), which means that Elkelesh et al. (2019) requires approximately 25 to 313 times more computations than our proposed method. The performance presented are for 75 and 150 iterations of the algorithm, representing around 70 and 140 times slower performance than our approach, respectively, while

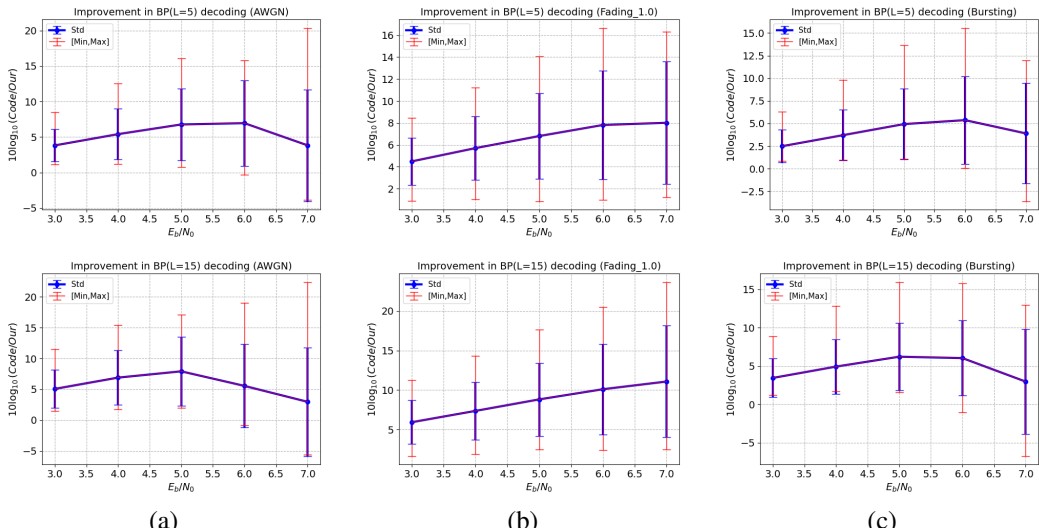

Figure 9: Statistics of improvement in dB for the (a) AWGN, (b) fading, and (c) bursting channel on the *sparse codes* only. We provide the mean and standard deviation as well as the minimum and maximum improvements.

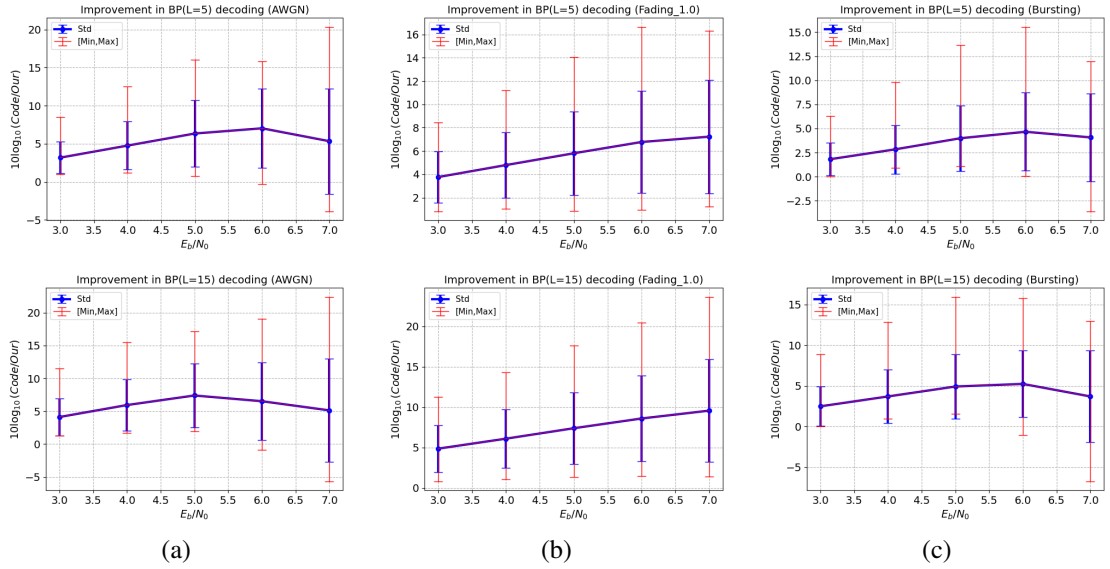

Figure 10: Statistics of improvement in dB for the (a) AWGN, (b) fading, and (c) bursting channel on all the codes from Table 3. We provide the mean and standard deviation as well as the minimum and maximum improvements.

they remain below our performance. We note here, as described in the paper, that combining the methods by allowing the perturbation of the parity-check matrix at a local minima may allow the discovery of other better local optimum.

# I  BER VS SNR CURVES

We provide standard visualizations of the BER and BLER performance with respect to the performance on multiple codes and channels. Figures 13, 14 provide performance on the AWGN and fading channel, respectively.

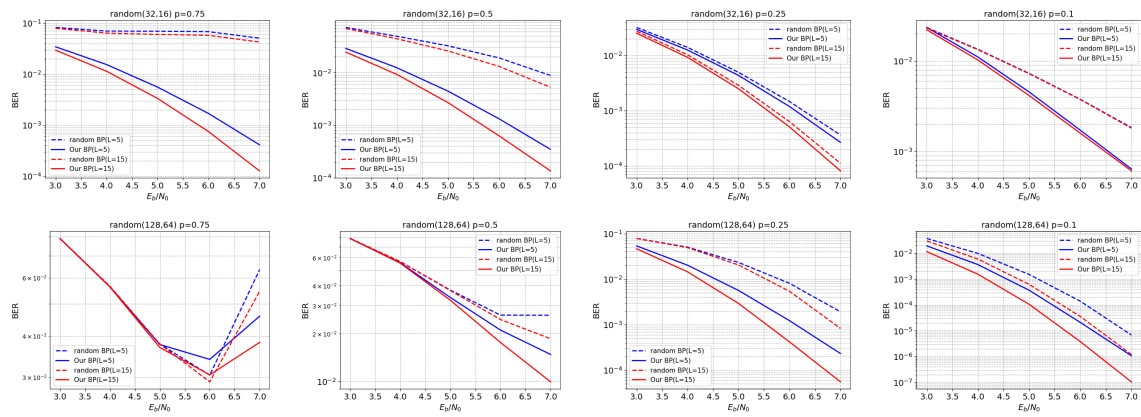

Figure 11: Performance of the method on random codes under different sparsity rate initialization $p$.

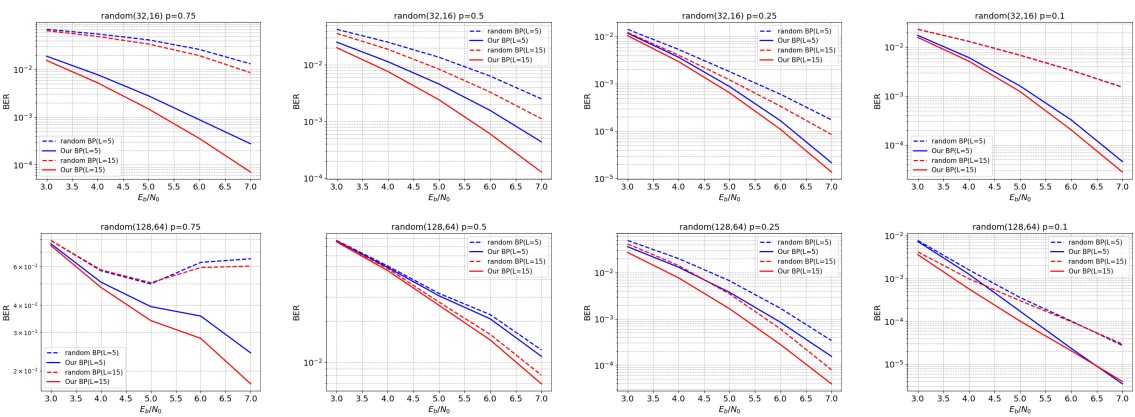

Figure 12: Performance of the method on constrained systematic random codes under different sparsity rate initialization $p$ on the AWGN channel.

## J    COLUMN WEIGHT DISTRIBUTION

We provide in Figure 15 an analysis of the column weight distribution of the original and learned parity check matrices on several codes and channel settings. While the method modifies substantially the distribution for non-sparse codes, BP's inductive bias seems to push the LDPC codes towards a non-uniform distribution of the variable nodes' degree.

## K    PERFORMANCE ON LARGER NUMBER OF BP ITERATIONS

We provide in Table 6 a comparison of the performance of the methods with larger number of BP iterations.

## L    VISUALIZATION OF THE LEARNED CODES

We present in Figure 16 the modification of the code through learning.

## M    TRANSFERABILITY OF THE LEARNED CODES BETWEEN CHANNELS

In classical error-correcting code design, the underlying channel model is typically assumed to be fixed, with theoretical derivations often relying on simplified channels like the Gaussian channel.

Table 5: A comparison of the negative natural logarithm of Bit Error Rate (BER) for several normalized SNR values of our method with classical codes. Higher is better. BP results are provided for 5 iterations in the first row and 15 in the second row. The best results are in bold. The training range is defined as $E_b/N_0 = \{5\}$. GA denotes the genetic algorithm of Elkelesh et al. (2019) with $k$ training iterations.

| Channel | AWGN | | | | | | | | | | | |
|---|---|---|---|---|---|---|---|---|---|---|---|---|
| Method | BP | | | Our | | | GA $k=75$ | | | GA $k=150$ | | |
| $E_b/N_0$ | 4 | 5 | 6 | 4 | 5 | 6 | 4 | 5 | 6 | 4 | 5 | 6 |
| CCSDS(128,64) | 6.46 | 9.61 | 13.99 | **7.34** | **10.48** | **14.37** | 6.86 | 9.95 | 13.38 | 7.09 | 10.40 | 14.08 |
| | 7.32 | 10.83 | 15.43 | **8.61** | **12.26** | 16.00 | 7.87 | 11.31 | 15.56 | 8.23 | 11.79 | **16.04** |

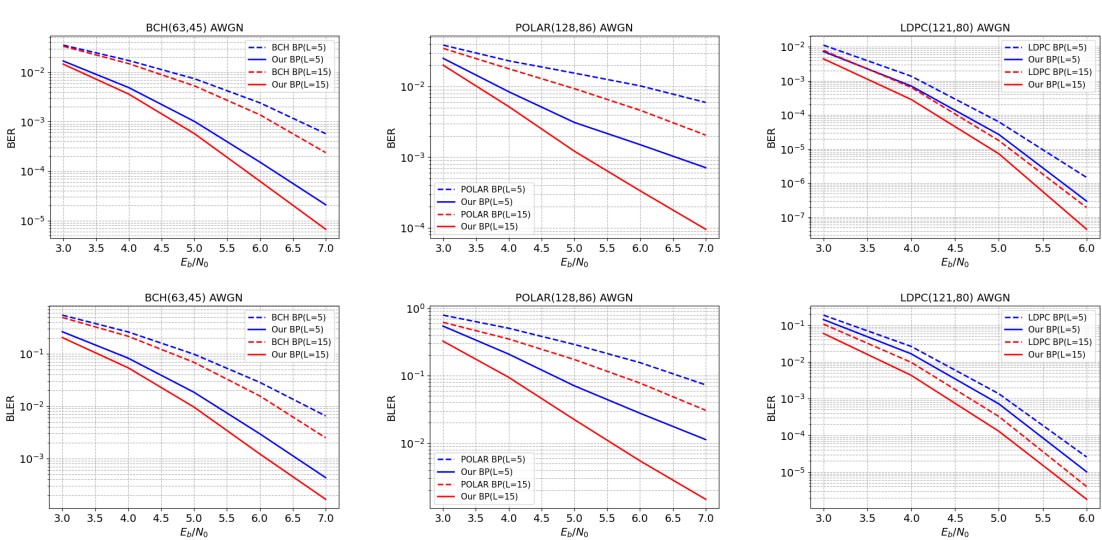

Figure 13: BER and BLER performance of the method on different codes on the AWGN channel.

We ran the transfer capabilities of three codes optimized for the AWGN channel to other channels. Orig. refers to the original performance of the codes optimized over the channel of interest, and transf. corresponds to the code optimized on AWGN and tested on the channel of interest. The learned codes seem to perform extremely well and clearly carry over different channels. This implies that learning the codes in the AWGN setting appears to facilitate convergence to better local minima compared to the (challenging) dedicated channels, enabling generalization to other codes.

## N   COMPARISON BETWEEN LS AND SGD

We provide in Table 8 a comparison of the performance between the line-search (LS) method and SGD. We optimized the same objective for 10K iterations using the Adam optimizer with three different learning rates ($10^{-4}, 10^{-3}, 10^{-2}$) and report the best result among them. The following table presents the negative natural logarithm of the Bit Error Rate ($-\ln(\text{BER})$, higher is better) for $E_b/N_0 \in \{4, 5, 6\}$ dB, with varying BP iterations ($L = 5$ and $L = 15$). As observed in the table, the gradient-based optimization (Adam) consistently yields lower scores compared to the Line-Search method. This indicates that standard gradient descent is prone to converging to poor local minima in the vicinity of the code initialization, whereas the Line-Search method effectively navigates the optimization landscape.

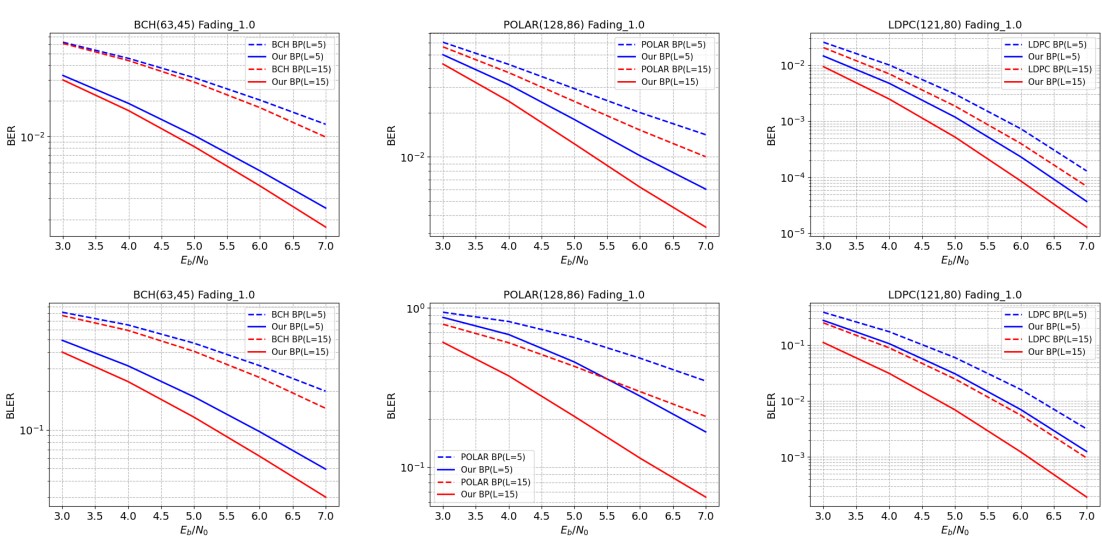

Figure 14: BER and BLER performance of the method on different codes on the Fading channel.

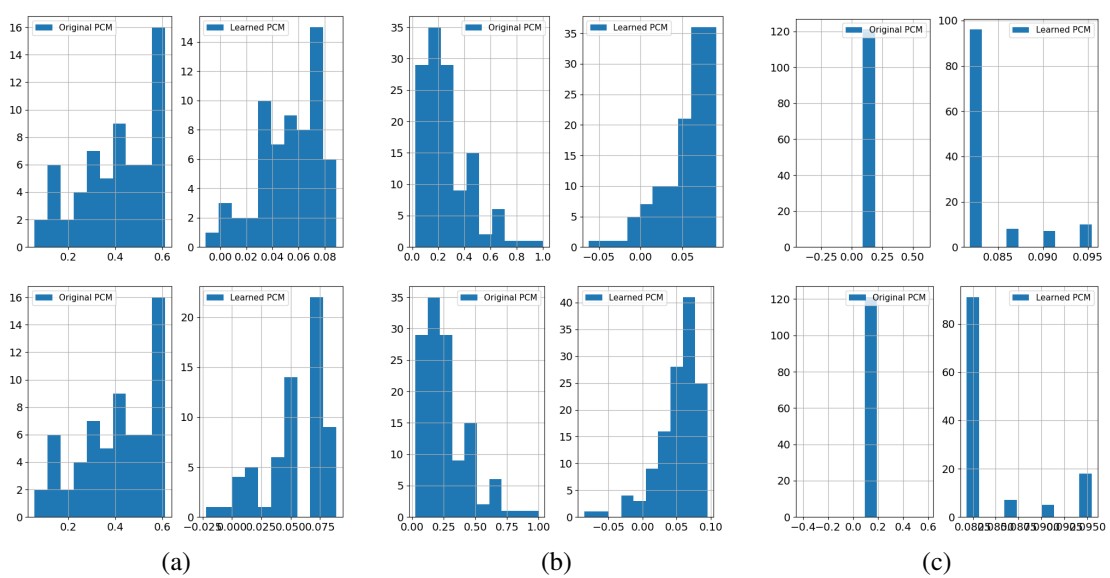

Figure 15: Column weight distribution of the parity check matrices on the (a) BCH(63,45) (b) POLAR(128,86) and (c) LDPC(121,80) codes. Top row and second row are for the AWGN and Fading channel, respectively.

Table 6: A comparison of the negative natural logarithm of Bit Error Rate (BER) for several normalized SNR values of our method with various number of iterations. Higher is better.

| Method | BP(L=5) | | | Our BP(L=5) | | | BP(L=50) | | | Our BP(L=50) | | | BP(L=100) | | | Our BP(L=100) | | |
|---|---|---|---|---|---|---|---|---|---|---|---|---|---|---|---|---|---|---|
| $E_b/N_0$ | 4 | 5 | 6 | 4 | 5 | 6 | 4 | 5 | 6 | 4 | 5 | 6 | 4 | 5 | 6 | 4 | 5 | 6 |
| BCH(63,45) | 4.06 | 4.91 | 6.04 | 5.44 | 6.93 | 8.60 | 4.34 | 5.56 | 7.30 | 5.83 | 7.59 | 9.41 | 4.44 | 5.93 | 7.96 | 5.86 | 7.64 | 9.44 |
| CCSDS(128,64) | 6.46 | 9.61 | 13.99 | 7.34 | 10.48 | 14.37 | 8.00 | 12.38 | 17.21 | 9.36 | 12.96 | 16.23 | 8.27 | 12.84 | 17.46 | 9.71 | 13.39 | 16.93 |
| LDPC(121,60) | 4.81 | 7.17 | 10.75 | 7.68 | 10.85 | 14.24 | 5.64 | 8.83 | 13.29 | 9.59 | 12.09 | 14.50 | 5.77 | 9.23 | 13.96 | 9.91 | 12.34 | 14.59 |
| POLAR(128,86) | 3.76 | 4.17 | 4.58 | 4.83 | 5.87 | 6.58 | 4.46 | 5.64 | 6.85 | 5.81 | 8.10 | 10.41 | 4.64 | 6.07 | 7.57 | 5.99 | 8.62 | 11.38 |

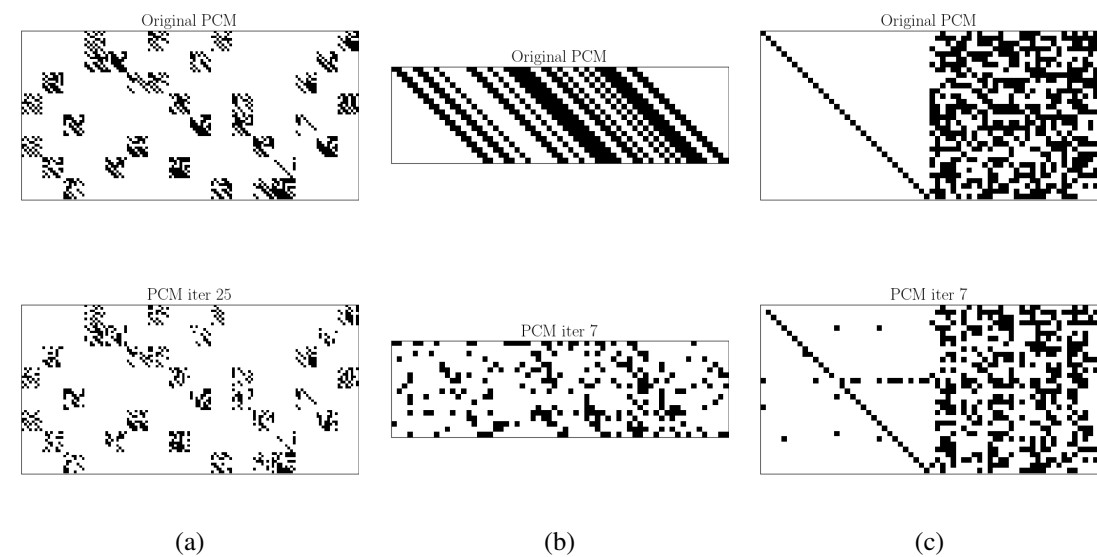

(a)                                    (b)                                    (c)

Figure 16: Visualization of the original (first row) and the learned parity check matrices (PCM; second row) for (a) LDPC(128,64), (b) BCH(63,45) and (c) Random(64,32,$p = 0.5$). "PCM iter X" = final optimization iteration parity check matrix.

Table 7: A comparison of the negative natural logarithm of Bit Error Rate (BER) for several normalized SNR values of our method optimized on a given channel and of our method optimized ont the AWGN channel and transfered to the channel of choice (e.g., AWGN → Fading) with various number of iterations. Higher is better.

| Channel | Fading | | | AWGN → Fading | | | Bursting | | | AWGN → Bursting | | |
|---|---|---|---|---|---|---|---|---|---|---|---|---|
| $E_b/N_0$ | 4 | 5 | 6 | 4 | 5 | 6 | 4 | 5 | 6 | 4 | 5 | 6 |
| BCH(63,45) | 3.96 | 4.58 | 5.27 | 3.94 | 4.53 | 5.17 | 4.05 | 5.07 | 6.27 | 4.69 | 5.93 | 7.30 |
| | 4.10 | 4.80 | 5.56 | 4.09 | 4.75 | 5.47 | 4.21 | 5.40 | 6.85 | 4.87 | 6.22 | 7.74 |
| LDPC(96,48) | 5.37 | 6.51 | 7.71 | 6.42 | 7.84 | 9.38 | 5.90 | 8.19 | 10.91 | 6.16 | 8.46 | 11.28 |
| | 6.14 | 7.38 | 8.65 | 7.53 | 9.16 | 10.75 | 6.71 | 9.28 | 11..75 | 7.05 | 9.63 | 12.53 |
| POLAR(128,86) | 3.64 | 4.28 | 4.94 | 3.46 | 3.98 | 4.52 | 3.69 | 4.51 | 5.18 | 4.16 | 5.24 | 6.18 |
| | 3.92 | 4.70 | 5.52 | 3.72 | 4.40 | 5.12 | 3.87 | 4.91 | 5.91 | 4.51 | 5.94 | 7.36 |

Table 8: A comparison of the negative natural logarithm of Bit Error Rate (BER) for several normalized SNR values of our line-search method (LS) compared to SGD. Higher is better.

| Channel | SGD | | | LS | | |
|---|---|---|---|---|---|---|
| $E_b/N_0$ | 4 | 5 | 6 | 4 | 5 | 6 |
| BCH(63,45) | 4.05 | 4.9 | 6.03 | 5.44 | 6.93 | 8.60 |
| | 4.19 | 5.23 | 6.59 | 5.70 | 7.35 | 9.16 |
| LDPC(121,60) | 4.81 | 7.18 | 10.67 | 7.70 | 10.87 | 14.25 |
| | 5.30 | 8.00 | 11.59 | 8.86 | 11.91 | 14.41 |
| POLAR(128,86) | 4.04 | 4.83 | 5.44 | 4.83 | 5.87 | 6.58 |
| | 4.23 | 5.23 | 6.11 | 5.37 | 6.88 | 8.10 |

