# OpenReview forum: "Factor Graph Optimization for Belief Propagation Decoding"
_ICLR.cc/2026/Conference — Submitted to ICLR 2026_

### Official Review · Reviewer_pZQ6 · 2025-10-17

**Soundness:** 2
**Presentation:** 2
**Contribution:** 2
**Rating:** 2
**Confidence:** 3

**Summary:**

This paper proposes a new framework for optimizing the design of error-definite codes using gradient methods. Conventional design methods for LDPC codes, etc., have fixed the code structure and optimized performance based on analytical and combinatorial techniques. However, in recent years, there has been a demand for flexible code designs that can adapt to new communication channels such as short block lengths, IoT, and 5G.
While many machine learning approaches have focused on designing "neural decoders," this research focuses instead on optimizing the "code itself (factor graph)."
The core idea of this research is, assuming a Belief Propagation (BP) decoder, to machine-learn the optimal code structure for the BP decoder.

**Strengths:**

Originality: This research is the first attempt to optimize a code structure (factor graph) using gradient descent while keeping the belief propagation decoder fixed.
Quality: As long as the authors examined, the superiority of the proposed method is shown objectively.
Clarity: What was done is clear.
Significance: Unlike conventional neural decoder-type ML-ECC, this method is compatible with existing BP implementations and has low implementation costs. The resulting code is sparse, practical, and performs well even in high SNR environments.

**Weaknesses:**

The relaxation method used for optimization has no theoretical basis. While significant performance improvements are observed for short and medium block lengths, this comparison is based solely on BP decoding. Unless the block length is sufficiently long, BP is strongly affected by cycles in the graph and is not necessarily a good decoding method. Performance comparisons with the current best codes for short and medium block lengths are desirable.

**Questions:**

How good is the performance obtained compared to the current best codes for short and medium block lengths?

---

> ### Author Response · Authors · 2025-11-19
>
> We thank the reviewer for their feedback.
>
> > ‘The relaxation method used for optimization has no theoretical basis.’
>
> - As noted in our manuscript (L242), we utilize the gradient estimation techniques analyzed in Bengio et al. (2013) and Yin et al. (2019). These works provide the theoretical justification for using surrogate gradients (like STE) to approximate gradients through discrete variables.
> - Our line-search approach is grounded in standard non-convex optimization theory under our binary constraints. It ensures monotonic descent of the loss function (or its surrogate), guaranteeing convergence to a local stationary point.
> We now provide in Appendix N a comparison with SGD optimization showing the large superiority of our approach.
> - This approach effectively bridges the gap between discrete coding constraints and differentiable optimization, which makes it a theoretically grounded approach.
>
> > ‘Unless the block length is sufficiently long, BP is strongly affected by cycles in the graph and is not necessarily a good decoding method.’’
>
> - We agree that standard codes suffer from loopy BP at short lengths due to cycles (absence of local tree-like connectivity). However, this is precisely the motivation for our work.
> - BP is the industry standard for high-throughput, low-latency decoding. Other decoders that handle short codes well (like SCL or MLD) are often too computationally expensive or serial for ultra-reliable low-latency communication scenarios and require the deployment of multiple decoders on the device. Our method explicitly optimizes the code structure to be robust to BP decoding even at short lengths.
>
> > ‘How good is the performance obtained compared to the current best codes for short and medium block lengths?’
>
> - The current SOTA codes in the short-length regime are Polar codes with Successive Cancellation List (SCL) decoding. As shown in Table 2, our learned codes (decoded via BP) outperform Polar+SCL. The learned codes obviously outperform the current best LDPC codes in this regime.
> - For medium/long-block length, 5G LDPC codes are state-of-the-art, our method cannot improve them, as they are either already close to the capacity or they define a very good local minima that first-order methods cannot overcome.

---

> > ### Comment · Reviewer_pZQ6 · 2025-11-26
> >
> > Thank you for your answer. I raised my evaluation from 2 to 4.

---

### Official Review · Reviewer_hwrJ · 2025-10-30

**Soundness:** 4
**Presentation:** 4
**Contribution:** 3
**Rating:** 8
**Confidence:** 4

**Summary:**

The paper suggests learning the Tanner graph of an error correcting code by optimizing the performance of a BP decoder on that code. In order to achieve this, the BP decoding is formulated in a way that is differentiable. Results show significant improvements in bit error rate (BER) as a result of the optimization.

**Strengths:**

The paper is well-written and easy to follow. I agree with the authors that it is better to use "data driven" approaches to optimize the code for BP rather than using a gigantic transformer to learn how to decode. The results show a clear advantage when the search is well initialized.

**Weaknesses:**

The work could be improved by showing the importance of a good initialization for the code optimization. For example, figure 2 shows that using the optimization method starting from a highly designed 5G code will lead to a better code. But what happens if we initialize with a bad code? Since the optimization landscape is highly nonconvex, I would presume that a good initialization is crucial.

The work could also be improved by comparing to other numerical methods for optimizing codes. Although the authors call their approach "data driven", I see it more as a competitor to methods such as density evolution and EXIT diagrams which have existed for over 20 years. In such methods, the expected BER of a code with certain properties is estimated and code optimization is performed by numerically optimizing these performance as a function of the parameters. I think both approaches have their strengths and weaknesses but the paper could be improved by discussing these explicitly.

**Questions:**

See the section on weakenesses.

---

> ### Author Response · Authors · 2025-11-19
>
> We thank the reviewer for the very supportive review and their precise understanding of our contributions.
>
> >The work could be improved by showing the importance of a good initialization for the code optimization
>
> We agree that initialization is crucial given the non-convex nature of the optimization landscape.
> To demonstrate the robustness of our method, we intentionally included results in Table 1 using dense or structurally distinct codes (e.g., BCH, Polar codes) as initializations. These represent challenging, suboptimal starting points compared to sparse LDPC structures, yet our method successfully optimizes them.
>
> We further corroborate this by explicitly quantifying the impact of initialization density in the ablation study presented in Figure 3.
>
> > The work could also be improved by comparing to other numerical methods for optimizing codes
>
> We thank the reviewer for the reference to other important related works.
>
> DE and EXIT methods estimate the expected performance of a code ensemble by tracking the evolution of statistical averages, such as the probability density functions of messages (DE) or the average mutual information (EXIT charts) as they pass through the iterative decoding process.
>
> A key feature is that these techniques are primarily designed for asymptotic analysis, assuming an infinite codeword length and an unlimited number of decoding iterations.
>
> Finally, the primary use is to numerically optimize macroscopic code parameters, such as the degree distributions, to maximize the iterative decoding threshold.
>
> Thus, DE/EXIT methods provide a theoretical upper bound on performance using approximations and asymptotic analysis, guiding the design of code ensembles. Conversely, factor graph optimization (ML-based) directly optimizes finite-length, practical code designs or decoder parameters for a specific channel and a limited number of iterations by training on empirical data, aiming for the best practical performance rather than the asymptotic theoretical limit.
>
> These points are now addressed in the revision.

---

> > ### Comment · Reviewer_hwrJ · 2025-11-25
> > **Thanks**
> >
> > Thank you for the comments. I would be happy to see this paper published.

---

### Official Review · Reviewer_26ev · 2025-11-01

**Soundness:** 3
**Presentation:** 2
**Contribution:** 2
**Rating:** 2
**Confidence:** 5

**Summary:**

This paper presents a gradient-based, data-driven method for designing sparse-graph codes tailored to belief propagation (BP) decoding. The main contribution is to learn the factor-graph structure through a differentiable representation that enables backpropagation. Specifically, the authors start from a complete bipartite graph where the edges are learnable.
The approach is interesting and, as the authors show, leads to codes that outperform some existing ones. However, I recommend rejecting the paper due to the limited scope of its contribution, the weak experimental comparisons, and the additional concerns detailed in the "Weaknesses" section.

**Strengths:**

The approach is interesting and, as the authors show, leads to codes that outperform some existing ones.

**Weaknesses:**

This paper was previously submitted to ICLR 2025 and rejected for well-founded reasons (see OpenReview). I would like to disclose that I served as one of the reviewers of that earlier submission. Upon inspection, this version appears nearly identical to the previous submission, with minimal or no substantive changes. Therefore, the same weaknesses identified in the earlier reviews remain unaddressed.
In short, the main issues are:

1. The writing of the paper requires significant improvement.
2. More importantly, the proposed codes do not outperform the state of the art. Yes, they do perform some good codes, but not the best ones.
3. While the work introduces some interesting ideas, it does not demonstrate sufficient impact or advancement to merit acceptance at a top-tier venue like ICLR. The contribution is more suitable for a specialized (not major) venue such as ISIT or ITW.

Overall, in my opinion does contain some interesting and potentially valuable ideas that deserve to be published. However, I also strongly believe that it does not merit publication in a top-tier conference/Journal. The decision made in the previous submission was well justified and should be upheld.

**Questions:**

I do not have additional questions beyond those raised in the previous submission. The paper is not technically flawed; however, its significance and potential impact are insufficient to merit publication in a venue of this caliber. In the opinion of this reviewer, the contribution is minor (besides the fact that the proposed approach does not surpass the state of the art)

---

> ### Author Response · Authors · 2025-11-19
>
> We thank the reviewer for their transparency regarding the previous submission.
>
>
> > ‘This paper was previously submitted to ICLR 2025 and rejected for well-founded reasons (see OpenReview). I would like to disclose that I served as one of the reviewers of that earlier submission’
>
> - We addressed all the reviewer’s questions and answered all of the reviewer’s concerns. No final answer was given by the reviewer in the previous submission.
>
> - We will be happy to discuss factual issues of the presentation, method and results.
>
> > ‘Upon inspection, this version appears nearly identical to the previous submission, with minimal or no substantive changes’
>
> We provide extended analysis compared to the original 2025 manuscript, including performance comparisons with 5G standards and comparison with SOTA Polar codes. We demonstrate the transferability of our codes between different channel models. We provide new results demonstrating performance stability with a large number of decoding iterations. Finally, we integrated all clarity-related feedback from previous reviewers.
>
> > ‘The writing of the paper requires significant improvement.’
>
> - We have proofread the manuscript.
> - As no other reviewers raised concerns regarding clarity or writing quality, we are unable to act on this feedback without specific examples. We remain open to correcting any specific issues pointed out.
>
> > ‘More importantly, the proposed codes do not outperform the state of the art. Yes, they do perform some good codes, but not the best ones.’
>
> - As shown in Table 2, our method outperforms SOTA Polar codes under SCL (L=32) decoding for short-length codes ($n=\{32, 64, 128\}$). They also certainly improve upon the best LDPC codes in the short-length regime.
> - Even where performance is "on-par" with SOTA, the significance is that an ML-based method generated codes matching the best theoretical codes developed over the last 70 years. This is the first time an ML approach for BP code design has achieved this, marking a distinct milestone in the intersection of Information Theory and Deep Learning.
>
> > ‘While the work introduces some interesting ideas, it does not demonstrate sufficient impact or advancement to merit acceptance at a top-tier venue like ICLR. The contribution is more suitable for a specialized (not major) venue such as ISIT or ITW.’
>
> - Historically, ICLR and similar top-tier ML venues have published numerous papers on Neural Error Correction.
> - Unlike previous works that suffered significant complexity degradation to apply ML, our method maintains theoretical rigor while utilizing modern machine learning techniques to reach SOTA performance while requiring zero modification of the algorithm. This application of AI for probabilistic graphical models and Scientific Discovery is a core theme of ICLR.
> This has been pointed out by other reviewers too.

---

### Official Review · Reviewer_iT7u · 2025-11-01

**Soundness:** 3
**Presentation:** 3
**Contribution:** 3
**Rating:** 4
**Confidence:** 3

**Summary:**

The paper proposes to directly learnthe factorgraph for BP by backpropagating through a tensorized BP and updating the binary parity-check matrix with a discrete-aware (line-search) step, showing better BP decoding on several codes/channels.

**Strengths:**

Keeps standard BP as the final decoder (more deployable with respect to other methods)

Differentiable formulation of BP over a learnable adjacency is novel to my knowledge

A clever and novel binary-aware update to make STE actually work is designed and tested

Consistent empirical gains across multiple code families

**Weaknesses:**

1. The paper sounds a bit like it is the first work to learn the factor graph. It would be good to have a comparison to PEG, differentiable LDPC search, neural BP with learnable edges, etc.

2. You start from a dense bipartite graph and run tensor BP; the training numbers are heavy for relatively small codes. Is it possible to use this on realistic blocklengths or 5G-like structured graphs? It would be good to quantify the cost, show a lighter variant (e.g. start from structured sparse, optimize only a subset), or show it works on a realistically constrained graph

3. You claim “factor graph optimization,” but you don’t show girth or cycle-spectrum improvements. It would be good to add before-after cycle histograms, degree constraints kept... Otherwise it seems difficult to distinguish the method from small edge changes driven by the loss (or be convincing that the latter is enough).

4. optimize for 5 iteration, test at 15 iterations could look a bit tuned to the experiment

5. Ablations on the key trick (binary-aware line search) are missing.

**Questions:**

Please address the weaknesses, I will consider increasing my score upon convincingly addressing them.

---

> ### Author Response · Authors · 2025-11-19
>
> We thank the reviewer for their comments and the time spent evaluating our work. We have addressed each point below.
>
> > ‘The paper sounds a bit like it is the first work to learn the factor graph. It would be good to have a comparison to PEG, differentiable LDPC search, neural BP with learnable edges, etc.’
>
> - We have compared our method against PEG on three different codes in Table 1, demonstrating superior performance.
> - To the best of our knowledge, existing Neural BP methods focus on learning weights for a fixed topology. In contrast, our method learns the topology (edges) itself. Extending our framework to include learnable non-binary weights would indeed bridge the gap to Neural BP, but our primary contribution remains structure learning.
> - Regarding differentiable LDPC search, we are eager to include relevant comparisons. If the reviewer could provide specific references to the intended literature, we will happily include a discussion or comparison in the final revision.
> - Regarding genetic algorithms, we provide a comparison with the data-driven method of Elkelesh et al. in Appendix H.
>
> > ‘You start from a dense bipartite graph and run tensor BP; the training numbers are heavy for relatively small codes. Is it possible to use this on realistic blocklengths or 5G-like structured graphs? It would be good to quantify the cost, show a lighter variant (e.g. start from structured sparse, optimize only a subset), or show it works on a realistically constrained graph’
>
> - While the general formulation allows starting from a dense bipartite graph, our method is fully compatible with sparse initializations. Thus, we don’t necessarily start from a dense bipartite graph. We emphasize that the choice of initialization is critical in this non-convex optimization landscape.
> - The training is fast while the large number of training samples is required to have a batch containing sufficient errors to sample the loss landscape. Following the reviewer recommendation (cf below), we now provide in Appendix N a comparison with SGD showing our method largely outperform classical small batch stochastic optimization.
> - We demonstrate scalability and applicability to realistic blocklengths in Table 1, where we apply our optimization starting from standard 5G-LDPC codes. In these experiments, the factor graph weighting is initialized using the sparse 5G structure rather than a dense graph. Initialization with other codes such as BCH codes are indeed dense.
> - We show the impact of sparsity of the initialization in the ablation study in lines 411-419 and figure 3.
>
> > ‘You claim “factor graph optimization,” but you don’t show girth or cycle-spectrum improvements. It would be good to add before-after cycle histograms, degree constraints kept... Otherwise it seems difficult to distinguish the method from small edge changes driven by the loss (or be convincing that the latter is enough).’
>
> - We have provided an analysis of the graph structural properties, including sparsity, PCM column weights, and girth, in lines 428-431.
> - Our observations indicate that while the global girth may not change, the optimization modifies the graph connectivity (Appendix L).
>
> > ‘optimize for 5 iterations, test at 15 iterations could look a bit tuned to the experiment’
>
> - We optimized for 5 iterations to manage computational resource constraints. However, testing at 15 iterations serves a dual purpose: it adheres to standard industrial practice for decoding, and more importantly, it demonstrates the generalization capability of the learned graph. The fact that a graph optimized for a 5-iteration unfolding performs well at 15 iterations indicates that the learned topology is robust and not overfit to the training depth.
> - Finally, we also provided in Table 6 (Appendix K, cf line 360) the performance of our method compared to the baseline for 50 and 100 iterations.

---

> ### Author Response · Authors · 2025-11-19
>
> > ‘Ablations on the key trick (binary-aware line search) are missing.’
>
> - An ablation removing the binary-aware line search essentially reduces the method to standard iterative SGD.
> - As detailed in lines 247-251, standard SGD oscillates and fails to converge in this discrete setting and typically becomes trapped in poor local extrema with performance similar to the initialization. Therefore, the line search is not merely a "trick" or hyperparameter to be tuned, but a fundamental mathematical requirement for the convergence of our discrete relaxation approach.
> Following the reviewer’s recommendation we provide an ablation table where we compare the performance of the line-search (LS) method with SGD.
> We optimized the objective for 10K iterations using the Adam optimizer with three different learning rates ($10^{-4}, 10^{-3}, 10^{-2}$) and report the best result among them. The following table presents the negative natural logarithm of the Bit Error Rate ($-\ln(\text{BER})$, higher is better) for $E_b/N_0 \in \{4, 5, 6\}$ dB, with varying BP iterations ($L=5$ and $L=15$).
>
> | Eb/N0         | SGD: L=5        | LS: L=5          | SGD: L=15       | LS: L=15         |
> |---------------|-----------------|------------------|-----------------|------------------|
> | BCH(63,45)    | 4.05,4.9,6.03   | 5.44,6.93,8.60   | 4.19,5.23,6.59  | 5.70,7.35,9.16   |
> | POLAR(128,86) | 4.04,4.83,5.44  | 4.83,5.87,6.58   | 4.23,5.23,6.11  | 5.37,6.88,8.10   |
> | LDPC(121,60)  | 4.81,7.18,10.67 | 7.70,10.87,14.25 | 5.30,8.00,11.59 | 8.86,11.91,14.41 |
>
> As observed in the table, the gradient-based optimization (Adam) consistently yields lower scores compared to the Line-Search method. This indicates that standard gradient descent is prone to converging to poor local minima in the vicinity of the code initialization, whereas the Line-Search method effectively navigates the optimization landscape.
>
> This ablation is added in the revised manuscript in Appendix N.

---

> > ### Comment · Reviewer_iT7u · 2025-11-23
> > **Thank you for the answers**
> >
> > It appears that the authors have addressed my main concerns, so I am inclined to raise my score. However, regarding their claims about SOTA performance, I think it is crucial to make very clear in which regime (e.g., short blocklengths, specific channel models, etc.) they actually match or surpass current best codes, and what the limitations are (e.g., no improvements for long 5G LDPC codes or when using stronger decoders). This is fundamental to understand and position their contribution, or whether the contribution "just" recovers SOTA in a smarter (learnable) way (if this is the case, do you get a clear win in some regime anyways? If this regime exists, is it relevant for applications?).

---

> > > ### Author Response · Authors · 2025-11-23
> > >
> > > We thank the reviewer for the important remarks and supportive comments.
> > >
> > > > regarding their claims about SOTA performance, I think it is crucial to make very clear in which regime (e.g., short blocklengths, specific channel models, etc.) they actually match or surpass current best codes, and what the limitations are (e.g., no improvements for long 5G LDPC codes or when using stronger decoders).
> > >
> > > - As is common with neural decoder works, we are indeed focusing on the short blocklength regime (cf, Abstract line 15 and lines 40-46, 82-86, 460-463).
> > > - The limitation of the method lies in its first-order formulation which restricts the information of descent by the gradient only (lines  460-463). Thus, large sparse codes close to the channel capacity already provide good local minima which requires non-convex optimization tools to extract from it (Appendix A).
> > > - Expanding our method with non-convex optimization tools may help discovering better large codes.
> > >
> > > We have updated the manuscript to make this distinction explicit in the motivation paragraph (lines 71-73):
> > > *The primary focus of our method is the short blocklength regime. In this setting, classical asymptotic constructions often fail to provide competitive performance, whereas our learned codes demonstrate a clear advantage over state-of-the-art solutions.*
> > >
> > >
> > > > This is fundamental to understand and position their contribution, or whether the contribution "just" recovers SOTA in a smarter (learnable) way (if this is the case, do you get a clear win in some regime anyways? If this regime exists, is it relevant for applications?).
> > >
> > > -  In the short blocklength regime the method is able to outperform SOTA codes (Table 2).
> > > - The method clearly allows the automatic learnable discovery of performing codes but is still constrained by the initialization as explained above.
> > > - This regime is of extreme importance in modern communication (lines 40-46, 82-86)

---

### Author Response · Authors · 2025-12-03
**Summary of Revisions and Responses to Reviewers**

We thank all the reviewers for their time and valuable feedback. In this general response, we summarize the main concerns raised by each reviewer and how we have addressed them in the revised manuscript and individual responses. We are encouraged that Reviewers iT7u, hwrJ, and pZQ6 have responded positively to our clarifications and revisions.
### Reviewer iT7u
- **Weakness Raised:** Request for comparisons to other methods (PEG, Neural BP) and 5G-like structures.
- **Our Answer:** We have comparisons to PEG (Table 1) showing superior performance and clarified the distinction from weight-based Neural BP. We also demonstrated superior performance in the short-length regime by initializing with 5G-LDPC codes.
- **Weakness Raised:** Lack of analysis on graph structural properties (girth, cycles).
- **Our Answer:** We have an analysis of structural properties (sparsity, PCM column weights, girth) within the paper and Appendix L, showing how optimization modifies connectivity while maintaining global girth.
- **Weakness Raised:** Missing ablation on the binary-aware line search (LS).
- **Our Answer:** We added a detailed ablation in Appendix N comparing our Line Search method against standard SGD (Adam). The results demonstrate that standard SGD fails to converge in this discrete setting, while our LS method consistently yields much lower BER, justifying its necessity.

**Status:** The reviewer acknowledged these updates and stated they are inclined to raise their score. We then refined the manuscript according to the reviewer’s final recommendation.

### Reviewer hwrJ
- **Weakness Raised:** Importance of initialization density and robustness to "bad" initialization.
- **Our Answer:** We provided an ablation study (Figure 3) quantifying the impact of initialization density. We also highlighted results in Table 1 where we successfully optimize starting from dense or suboptimal codes (BCH, Polar), demonstrating robustness.
- **Weakness Raised:** Comparison to numerical methods like Density Evolution (DE) and EXIT charts.
- **Our Answer:** We clarified the distinction in the revision between asymptotic theoretical bounds (DE/EXIT) and our finite-length, data-driven optimization. Our method optimizes specific instances for practical performance rather than ensemble averages.

**Status:** The reviewer reaffirmed their positive assessment (Rating: 8) and supports publication.

### Reviewer pZQ6

- **Weakness Raised:** Theoretical basis for the relaxation method and the validity of BP for short block lengths due to cycles.
- **Our Answer:** We referenced the theoretical grounding for surrogate gradients (Bengio et al., 2013; Yin et al., 2019) and the non-convex convergence guarantees of our line search. We clarified that our method explicitly optimizes the graph to be robust to cycles in the short-length regime, where BP is preferred for low latency.
- **Weakness Raised:** Performance relative to SOTA short codes.
- **Our Answer:** We pointed to Table 2, showing our learned codes outperform SOTA Polar codes with SCL decoding in the short-length regime ($N\leq128$), validating the method's practical contribution.

**Status:** The reviewer accepted these arguments and raised their score.

### Reviewer 26ev

- **Weakness Raised:** Concerns regarding novelty compared to a previous submission and the claim that codes do not outperform SOTA.
- **Our Answer:** We respectfully disagree regarding the "minimal changes." The current manuscript includes new extensive analysis on 5G/Polar codes, transferability, and stability, addressing previous concerns. Regarding SOTA, we have empirically demonstrated (Table 2) that our codes outperform the best Polar+SCL codes in the short-blocklength regime which is the focus of our paper.
- **Weakness Raised:** Fit for venue (ICLR vs. ISIT).
- **Our Answer:** As noted by other reviewers, this work applies modern ML techniques (gradient-based structure learning) to a classical problem without altering the inference algorithm. This intersection of "AI for Science" and probabilistic graphical models is well within the scope of ICLR.

**Status:** No action point has been given in the review and no update/answer has been made during the (previous) rebuttal period.

### **Summary**

We have addressed the technical concerns regarding comparisons, graph properties, and optimization baselines (ablation added). The consensus among Reviewers iT7u, hwrJ, and pZQ6 is that the method is sound and novel, and the results demonstrate a clear advantage in the targeted short blocklength regime.

---

### Meta-Review · Area_Chair_zPMm · 2026-01-03

**Summary:**

Most reviewers agree that the main idea of the paper is interesting and novel. Keeping the standard BP decoder and optimizing the factor graph with gradient descent is a clever and elegant idea.

However, several concerns remain. The impact of the results is limited. The improvements are mainly in the ultra-short blocklength regime. For the 5G LDPC (128,64) code, no improvement is shown. If the paper focuses on ultra-short lengths, it should better discuss and compare with stronger alternatives for this regime, such as generalized LDPC codes, non-binary LDPC codes, or quasi-cyclic codes, which are being considered for future communications beyond 5G.

The paper claims it can handle arbitrary modulations, but this seems to rely on bit-wise LLRs at the input. To naturally support arbitrary modulations, non-binary codes would be needed, but these are not considered.

There are also concerns about the discussion of Density Evolution and EXIT charts. The rebuttal mainly states that these tools are asymptotic, but they are known to relate to finite-length performance in the waterfall region through scaling laws, for example as shown in “Finitelength scaling for iteratively decoded LDPC ensembles,” IEEE Transactions on Information Theory, vol. 55, no. 2, pp. 473–498, Feb. 2009. I do not think this point, raised by reviewer hwrJ, is fully addressed.

Finally, the paper claims that optimization can be done using only the all-zero codeword due to symmetry. While this is standard for fixed codes, it is unclear if this still holds under the continuous relaxation used for gradient-based optimization. This likely needs a clearer explanation or formal proof.

Overall, while the idea is promising and the paper has improved since the previous submission, the current results and analysis are not strong enough for acceptance at ICLR.

**Reviewer Concerns:**

**Addressed**
- Clarified that the main focus of the paper is the short and ultra-short blocklength regime.
- Added explanations and ablations on initialization density and robustness to poor initial graphs.
- Added comparisons with PEG constructions and clarified differences with Neural BP.
- Added ablation studies for the binary-aware line search optimization method.
- Improved clarity on graph properties such as sparsity and girth.

**Still outstanding**
- Limited impact of results beyond ultra-short blocklengths, with no improvement shown for codes such as  5G LDPC (128,64).
- Missing comparison with stronger short-blocklength alternatives, such as generalized LDPC codes, non-binary LDPC codes, and quasi-cyclic codes.
- Claim of handling arbitrary modulations is not fully supported without non-binary codes.
- Discussion of Density Evolution and EXIT charts remains incomplete, ignoring known finite-length scaling results.
- Lack of formal justification that optimizing only the all-zero codeword remains valid under continuous relaxation.
- Overall impact and relevance for practical systems remain unclear.

**Reviewer Scores:**

**Reviewer Scores (Post-discussion estimate)**

- **Reviewer iT7u**
  Likely to slightly increase the score, as most of their concerns were addressed. 4--> 6

- **Reviewer hwrJ**
  Unlikely to change the score, as initial score was high (8).

- **Reviewer pZQ6**
Reviewer already acknowledge to raise from 2 to 4.

- **Reviewer 26ev**
  Unlikely to change the score, as concerns about impact and novelty remain. 2

Average rating is 5.

---

### Decision · Program_Chairs · 2026-01-26

Reject